# Recent Mechanisms of Neurodegeneration and Photobiomodulation in the Context of Alzheimer’s Disease

**DOI:** 10.3390/ijms24119272

**Published:** 2023-05-25

**Authors:** Matthew Su, Damir Nizamutdinov, Hanli Liu, Jason H. Huang

**Affiliations:** 1Department of BioSciences, Rice University, Houston, TX 77005, USA; 2Department of Neurosurgery, College of Medicine, Texas A&M University, Temple, TX 76508, USA; 3Department of Neurosurgery, Neuroscience Institute, Baylor Scott and White Health, Temple, TX 76508, USA; 4Department of Bioengineering, The University of Texas at Arlington, Arlington, TX 76010, USA

**Keywords:** Alzheimer’s disease, neurodegeneration, mechanisms, photobiomodulation, near-infrared light, transcranial

## Abstract

Alzheimer’s disease (AD) is a neurodegenerative disease and the world’s primary cause of dementia, a condition characterized by significant progressive declines in memory and intellectual capacities. While dementia is the main symptom of Alzheimer’s, the disease presents with many other debilitating symptoms, and currently, there is no known treatment exists to stop its irreversible progression or cure the disease. Photobiomodulation has emerged as a very promising treatment for improving brain function, using light in the range from red to the near-infrared spectrum depending on the application, tissue penetration, and density of the target area. The goal of this comprehensive review is to discuss the most recent achievements in and mechanisms of AD pathogenesis with respect to neurodegeneration. It also provides an overview of the mechanisms of photobiomodulation associated with AD pathology and the benefits of transcranial near-infrared light treatment as a potential therapeutic solution. This review also discusses the older reports and hypotheses associated with the development of AD, as well as some other approved AD drugs.

## 1. Introduction

Alzheimer’s disease (AD) is the world’s primary cause of dementia, a condition characterized by significant progressive declines in memory and intellectual capacities [1]. It is estimated that about 6.5 million Americans aged 65 years or older live with Alzheimer’s, which is predicted to more than double by 2060 [2]. AD can also be seen in patients younger than 65. Known as early-onset AD, this rarer form of the disease affects between 220,000 to 640,000 Americans [3]. Alzheimer’s has consistently appeared in the top ten causes of death in the United States, ranking at the 6th spot in 2019 and the 7th spot in 2020 and 2021. However, while other leading causes of death, such as stroke, HIV, and cardiovascular disease, have resulted in lower mortality in the past two decades, deaths due to AD have risen alarmingly by 145% [2].

While dementia is the main symptom of Alzheimer’s, the disease presents with many other debilitating symptoms. Alois Alzheimer, the first person to ever document this affliction, noted confusion, delusion, and behavioral changes in a 51-year-old woman named Auguste Deter [4]. Patients who begin to experience mild cognitive impairment due to Alzheimer’s will typically have difficulty recalling conversations they have had earlier in the day, navigating familiar areas, remembering words, and completing normal daily tasks [5,6]. At this point, the patient can still live and engage in activities independently despite their decreased mental functioning. Eventually, as more parts of the brain deteriorate and cognitive impairments worsen, the patient will start showing signs of Alzheimer’s dementia, causing them to lose a greater degree of independence. In the late stages of the disease, the patient will likely require care and attention at all times, unable to perform almost any tasks of daily living without assistance [5]. Currently, no known treatment can stop the irreversible progression of the symptoms.

Understanding the pathogenesis of this incurable disease is, therefore, more important than ever. Alzheimer’s is categorized as a neurodegenerative disease, and causes progressive damage to and death of the neurons in the brain [7]. The signature biomolecular hallmarks of AD are an extracellular aggregation of amyloid-beta (Aβ) plaques and intracellular aggregation of neurofibrillary tangles (NFTs) [7]. These harmful accumulative lesions lead to neural and synaptic degeneration [1,7].

In the past three decades, the amyloid hypothesis was the most accepted model for explaining the pathogenesis of AD. This hypothesis postulates that the buildup of Aβ plaques is the first in a cascade of events that results in neurodegeneration [8,9,10]. Aβ plaques derive from an altered amyloidogenic pathway of cleaving amyloid precursor protein (APP), an integral membrane protein [1,11,12]. In the physiologic non-amyloidogenic pathway that does not lead to amyloid plaque accumulation, APP is cleaved by the α-secretase enzyme at residues 16–17 of the Aβ domain and then the γ-secretase enzymes, producing a small, soluble hydrophobic fragment, p3, and a large, soluble ectodomain, APPsα. Prior research has shown that injecting APP ectodomains into the cerebrum boosts cognitive abilities and synaptic health, suggesting that the non-amyloidogenic pathway of APP has some neuroprotective qualities [7,13,14]. However, when a mutation occurs in the APP gene, the presenilin 1 or presenilin 2 genes, or the apolipoprotein E (APOE4) gene (Table 1) [15,16,17], the amyloidogenic pathway can ensue, where APP is first cleaved by β-secretase, yielding fragments of 89 to 99 amino acids bound by the C terminal to the plasma membrane [7,9,18,19]. The fragments are then cleaved by γ-secretase into Aβ1-40 and Aβ1-42 isoforms, the former being normal and soluble while the latter is insoluble and plaque-inducing. In the amyloidogenic pathway, the ratio of Aβ1-42:Aβ1-40 is dangerously high. The amyloid plaque buildup leads to an immune response involving a release of cytokines and microglia activation, creating a neurotoxic environment that causes cell death [7,9].

The Aβ plaque accumulation appears to be upstream to the formation of neurofibrillary tangles (NFT) [8]. These straight and fibrillary tangles consist of microtubule-associated tau protein, which normally binds with tubulin to make and stabilize networks of microtubules. Tau appears as six isoforms and is named depending on the amount of microtubule-binding domains. 3R and 4R correspond to three and four microtubule-binding domains, respectively, representing the primary tau species in Alzheimer’s pathology [10,20,21,22]. An environment with abundant extracellular Aβ activates various kinases, such as glycogen synthase kinase 3 (GSK3β) and cyclin-dependent kinase 5 (CDK5), to hyperphosphorylate tau, causing the protein to oligomerize. The microtubules become unstable and prone to dissociating into large tau filaments that can aggregate into insoluble NFTs in the neural cytoplasm. Intracellular NFTs contribute to poor neural signaling and apoptosis [7,23,24].

Despite the popularity of the amyloid hypothesis, modern research has compelled a good portion of the scientific community to reconsider the theory. Past studies looking at the effects of Aβ-targeting drugs have demonstrated that decreased levels of amyloid plaques failed to alleviate Alzheimer’s symptoms [25,26,27,28,29]. Additionally, many experiments have failed to not find a link between the presence of Aβ aggregation and cytotoxicity for neurons [25,30,31]. Recent amyloid imaging revealed that elderly patients without dementia may have similar levels of Aβ plaque as those with Alzheimer’s [25,32,33]. This suggests that amyloid buildup could be more of a result of aging. An alternative hypothesis that recently came to light is the tau hypothesis, which deems tau as the primary cause of AD as opposed to amyloid aggregation. Tau protein has been found to be strongly correlated with cell death and neurodegeneration, as well as clinical symptoms of AD [25,34]. APP C-terminal fragments could serve as the triggers for tau accumulation [25,35], and both entities could compound one another’s effect in disrupting neural synaptic function [25,36].

Furthermore, synaptic degeneration provides a pathological mechanism of Alzheimer’s that has recently attracted much attention as another possibility for the amyloid hypothesis. Dysfunctional changes to neuronal synapses could occur early on, before clinical symptoms of AD emerge, and some research has even suggested that these changes transpire before the signature markers of amyloid plaque and NFTs become evident (Figure 1) [8,9,10,15,16,22,27,32,34,37,38,39,40,41]. Although the factors that lead to these harmful synaptic alterations are not fully understood, the current research postulates that Aβ, tau, genetic risk factor Apolipoprotein E (ApoE4), and microglia all play major roles [38]. Aβ and tau protein could induce changes in the number, size, shape, and composition of synapses, resulting in cognitive deficiencies [38,39,40]. Synapse loss and dysfunction could grow more pronounced with the inheriting of the ApoE4 gene (Table 1), which can cause neural glial cells to prune synapses abnormally [38]. One such glial cell, microglia, normally functions in a neuroprotective role and can accumulate excessively in Alzheimer’s pathology, making neuroinflammation another factor worth considering (Figure 1) [38]. Thus, we are in a crucial stage of AD research, as more evidence is pointing to a complex pathology involving interactions between multiple factors as opposed to past models singling out certain common biomarkers that can trigger a cascade of damaging events (Table 2). Many of the listed biomarkers are used as AD diagnostic tools, and can be measured in the cerebrospinal fluid (CSF), plasma, or other bodily fluids depending on the sensitivity of the method used, the concentration of the targeted biomarker in the system, or the stage of the disease. Some biomarkers have been linked to the pathogenesis of AD which can be detected decades before the first clinical onset of AD (preclinical stage of disease) [42,43]. Table 2 summarizes the biomarkers linked to AD due to the excessive or toxic nature of the accumulated protein(s) associated with different mechanisms of AD pathogenesis, which will be discussed in this review. Thus, the most commonly reported and studied biomarkers can be sorted by subgroups based on their mechanisms of pathogenesis. For example, the accumulation of Aβ is linked with the APP and Aβ42/Aβ40 ratio; the Tau-related biomarkers are T-tau and P-tau; the neural damage-associated biomarkers are NfL and S100β; the biomarkers associated with neuroinflammation are GFAP, TNF-α, and IL-β; the synaptic biomarkers are α-synuclein and neurogranin; and the metabolic biomarkers are ApoE and GDNF [42,43].

In recent years, photobiomodulation (PBM) has emerged as an extremely promising treatment for improving brain function, especially after damage due to stroke, traumatic brain injury (TBI), neurodegenerative disorders, etc. Also known as low-intensity light therapy, this technique uses light in the red to near-infrared (NIR) range (600–1100 nm) [44,45]. The first in vivo experiment to demonstrate the cognitive benefits of PBM therapy was conducted in rabbits with small clot embolic strokes. The results showed significant behavioral improvements if the laser treatment was carried out within 6 h of the stroke [46]. PBM primarily works by boosting cell energy production and metabolism [44]. Two pathways behind this effect were elucidated, both involving the excitation of mitochondrial cytochrome c oxidase (CCO), the terminal enzyme in the electron transport chain (ETC) responsible for transferring electrons from cytochrome c to O_2_ [47,48]. The first-studied pathway shone a light on CCOs. When the electrons in the metal centers of CCO are excited by photon absorption (Figure 2) [44,45,47,48,49,50,51,52], nitrous oxide (NO) from CCO’s binuclear center (heme a3/CuB) is photodissociated. Decreasing amounts of NO, a known electron transport inhibitor in the ETC, raise the mitochondrial membrane potential (MMP), consequently increasing the proton gradient and ATP production [52]. This upregulates the function of reactive oxygen species (ROS) and calcium ions as secondary messengers, resulting in the activation of transcription factors and cell proliferation signaling molecules such as nuclear factor kappa-light-chain-enhancer of activated B cells (NF-kB) (Figure 2) [53,54]. Recently, research has demonstrated that light in the 900–1100 nm range can also produce similar increases in MMP and ATP production, albeit through a different pathway. In this second pathway, light appears to be absorbed by temperature-gated calcium channels, causing an increase in cytosolic calcium but a decrease in mitochondrial calcium [48].

The effects of PBM provide a host of neurobiological benefits. Because neural tissue is rich in and dependent on mitochondria [55] and many brain disorders are caused by mitochondrial dysfunction (e.g., decreased MMP, less ATP synthesis, etc.) [56], photobiomodulation addresses a key source of neuronal deterioration by boosting energy production. PBM delivered at appropriate levels generates low levels of ROS to promote cell proliferation and homeostasis while minimizing the mitochondrial oxidative stress that results from excessive ROS [57,58]. Low-intensity NIR light can additionally modulate the expression of pro-inflammatory cytokines by inhibiting NF-kB pathways and attenuating harmful neuroinflammation [59,60]. Impaired cerebral blood flow (CBF) is another chief cause of brain damage in neurodegeneration, but the dissociation of nitrous oxide induced by transcranial NIR light can improve blood circulation and oxygenation (Figure 1 and Figure 2), as NO is a powerful vasodilator [61]. Furthermore, many studies have suggested that the enhanced mitochondrial function brought about by PBM treatment reduces apoptosis [62,63], while the activation of neurotrophic factors such as brain-derived neurotrophic factor (BDNF) and neuronal growth factor (NGF) by PBM augment neurogenesis [64].

Today, PBM’s neurotherapeutic effects have been tested in various conditions, one of which is AD, the world’s leading cause of dementia. Nizamutdinov et al. conducted a study examining the effect of administering transcranial NIR light for 6 min twice a day for 8 continuous weeks on patients with dementia [65]. Using various neuropsychological tests to assess behavior, mood, and cognitive performance, they found that patients undergoing PBM treatment reported better levels of sleep and improved mood and energy [65]. One possible explanation for NIR light’s effectiveness in treating Alzheimer’s is its ability to improve CBF [66]. Hypertension is a risk factor for AD [37,67], and drugs targeting hypertension can decrease the incidence of Alzheimer’s [68]. High blood pressure can damage the vascular system in the brain, interfering with the function of the blood–brain barrier (BBB) and causing neuronal dysfunction [69,70]. As mentioned previously, the release of NO induced by PBM serves as a vasodilator, increasing cerebral blood flow and oxygenation to brain tissue [61]. Research has suggested that PBM therapy can support capillary-like structure formation by stimulating vascular endothelial growth factors [71,72]. Therefore, by lowering blood pressure, PBM offers not only a potential therapeutic approach for AD that slows down the progression of symptoms, but also a preventative treatment that regulates a strong risk factor of AD [66].

## 2. Mechanisms of Alzheimer’s Disease Pathogenesis

### 2.1. Role of Oxidative Stress

Oxidative stress is an imbalance between the production of reactive oxygen species (ROS) and the biological organism’s ability to counteract their effects [73]. ROS, as the name suggests, are derived from oxygen (e.g., superoxide radicals (O_2_-), hydrogen peroxide (H_2_O_2_), and hydroxyl radicals (HO-)) [73]. Superoxide radicals are normally produced by mitochondria from the ETC during cellular respiration, after which they can be converted to H_2_O_2_ by an enzyme called superoxide dismutase. Hydrogen peroxide can be further broken down to hydroxyl radicals, the most reactive of the free radical species, via the Fenton reaction [73,74,75]. In normal, healthy amounts, ROS serves numerous physiological functions, such as cell signaling and immune defense against pathogens [73,76,77]. However, ROS can over-accumulate and oxidize macromolecules such as membrane lipids, proteins, and nucleic acids, interfering with their essential function for cell survival and causing oxidative stress [78].

Because ROS can wreak havoc at such a fundamental cellular level, it should be no surprise that oxidative stress has been widely explored in various disease pathologies, and has been shown to play a fundamental role in neurodegenerative processes. The nervous system is particularly susceptible to oxidative stress damage, as the brain consumes 20% of the body’s supply of oxygen [79] and neurons do not undergo mitosis, allowing mitochondrial dysfunction to continue worsening with time [80]. While increased ROS production and mitochondrial oxidative stress naturally occur during the aging process [80], in AD, concentrations build up to neurotoxic levels thanks to amyloid-beta plaque, which is capable of complexing with metal ions such as copper, zinc, and iron as well as generating superoxides and hydrogen peroxide [81,82]. Two additional AD characteristics, DNA damage and harmful epigenetic modifications, have been linked to oxidative stress. The initial damage to nucleic acids can activate kinase enzymes and poly-ADP ribose polymerase, reducing NAD^+^ levels and increasing mitochondrial coupling to compensate. This ultimately results in increased ROS production (Figure 1) and increased DNA damage [83,84]. Oxidative stress can alter post-translational histone acetylation and methylation and disrupt normal gene expression as well. Specifically, H_2_O_2_ hypomethylates and increases the expression of the APP and β-secretase enzymes, accelerating Aβ formation [85].

Oxidative stress has been linked closely with Aβ plaque in Alzheimer’s pathology. More studies implicated oxidative stress as the primary villain in the amyloid hypothesis, as opposed to a by-product of Aβ aggregation [86]. Pro-oxidants were shown to lower the levels of APP produced from the non-amyloidogenic pathway in medium spiny neurons (MSN) [87]. In human neuroblastoma cells, pro-oxidants decreased α-secretase enzyme expression while upregulating β-secretase enzyme expression, which similarly favors the amyloidogenic pathway [87]. Interestingly, oxidative stress also plays a significant role in the tau hypothesis, suggesting it is a common denominator in different theories of AD pathogenesis [86]. ROS can disturb the physiological function of tau and stimulate GSK3β and other protein kinases, causing tau to become hyperphosphorylated [88]. Hyperphosphorylated tau protein loses the ability to bind to microtubules and instead oligomerizes into insoluble NFTs, which can, in turn, promote increased ROS generation and aggravate the effects of oxidative stress [89].

The abundance of research implicating oxidative stress in cognitive aging and neurodegeneration, even in divergent models of how AD arises, consists of driven efforts to examine the potential of antioxidant therapies [89,90]. A longitudinal study conducted in humans found that the consumption of vitamin E, an antioxidant, through foods and supplements correlated with less cognitive decline [91]. Unfortunately, using antioxidants to treat AD in clinical trials produced mixed results. For example, one study concluded that there was no decrease in the incidence of AD after several years of daily vitamin E consumption [92], and another study determined that daily alpha-tocopherol was able to slow down cognitive deterioration in AD patients [93]. Antioxidant therapy was criticized for not being able to specifically target sources of oxidative stress through simply ingesting foods and/or supplements [92].

Photobiomodulation provides an alternative and possibly more effective method for minimizing oxidative stress. Previous literature has shown that PBM causes the dissociation of nitrous oxide from Complex IV (cytochrome c oxidase, CCO) in the mitochondrial electron transport chain (Figure 2) [94]. NO normally inhibits the activity of Complex IV by binding to its heme α3 and CuB centers [94,95], so NIR light-induced dissociation upregulates CCO, strengthening the flow of electrons through the ETC; the proton gradient; and, subsequently, ATP production [94]. This boost in mitochondrial function corresponded to an increase in the mitochondrial membrane potential, leading to a transient increase in ROS generation and activation of NF-kB in the cytoplasm [57]. NF-kB was then able to exert its cytoprotective effects by moving to the nucleus and expressing various genes for the purpose of modulating oxidative stress and neuroinflammation [96]. It is quite interesting to note that an initial short burst of ROS production created by PBM resulted in the attenuation of future mitochondrial dysfunction and excessive ROS levels [97].

It, therefore, follows that when treating neural tissue with PBM therapy, parameters of light exposure need to be carefully optimized so as not to overdo the extra ROS generation, but still to allow for PBM’s beneficial effects. A study by Amaroli et al. in 2021 found that NIR light at a 980 nm wavelength shone at 0.8 W for 60 s resulted in better ATP synthesis when compared to lower power levels [98]. Lower power ranges (0.1–0.2 W) inhibited energy production. However, augmentation of oxidative stress was observed at 0.8 W power and even at lower power levels. This can be attributed to increased oxygen consumption and experimental conditions, as PBM was conducted on mitochondria in isolation, which might have caused the loss of some antioxidant molecules responsible for detoxifying ROS [98]. It is plausible that a high enough degree of oxidative stress augmentation could surpass what normal tissue levels of antioxidants can handle. Other research illustrated that the impact of PBM could be dose-, time-, and tissue-dependent. In 2019, the effects of PBM were assessed on the mitochondria of brain, muscle, and C6 astroglioma cells [99]. After exposing the cells to varying intensities of 10, 30, and 60 J/cm^2^ and measuring the effects 5 and 60 min post-irradiation, the results demonstrated that the three kinds of cells responded differently to the tested intensities. The brain mitochondria only saw an increase in complex IV activity 5 min after irradiation, but after 1 h, complex II activity was elevated to 60 J/cm^2^, while complex IV activity increased at all intensities [99].

PBM constitutes a very promising treatment because it specifically addresses a primary source of brain disease development: mitochondrial dysfunction. Future studies should further elucidate the precise mechanisms of PBM’s effects on mitochondria and the optimal parameters (e.g., wavelength, dose, timeframe) for low-intensity NIR light for enhancing mitochondrial activity.

### 2.2. Role of Neuroinflammation

Neuroinflammation is defined as an inflammatory response that occurs in the central nervous system (CNS), which includes the brain and the spinal cord [100]. This process entails the release of pro-inflammatory cytokines such as IL-1 (beta), IL-6, IL-18, and tumor necrosis factor (TNF); chemokines; small secondary messengers; and ROS, and is primarily mediated by CNS glial cells involved in the innate immune system response (i.e., microglia and astrocytes) [100,101]. When it occurs at appropriate levels, neuroinflammation provides numerous benefits to the host organism. Cytokines can play an important role in learning and memory (neuroplasticity), and can promote neural recovery after the CNS sustains an injury [100]. However, chronically high levels of neuroinflammation are characterized by the overproduction of cytokines and ROS, leading to neural cell death and contributing to neurodegeneration [100].

An imbalance in the innate immune system response can result in detrimental neuroinflammation. Cells such as microglia and astrocytes contain pattern recognition receptors (PRRs) that detect damage- or pathogen-associated molecular patterns (DAMPs or PAMPs) [101,102]. PRRs can be categorized into five families: toll-like receptors (TLRs), nucleotide-binding oligomerization domain-like receptors (NLRs), retinoic acid-inducible gene-I (RIG-I)-like receptors (RLRs), C-type lectin receptors (CLRs), and absent in melanoma-2 (AIM2)-like receptors (ALRs) [102,103]. Aβ and NFTs activate PRRs excessively, causing microglia and astrocytes to release cytokines and chemokines [104]. While these modulators of neuroinflammation are beneficial in low quantities and can reduce amyloid plaque [105], prolonged activation can lead to a cytokine storm, wherein a large number of cytokines, especially tumor necrosis factor alpha (TNF-α), can induce inflammatory cell death [106,107]. Elevated production of chemokines is also observed in microglia and astrocytes near Aβ plaque, and can exacerbate local neuroinflammation by recruiting more reactive glial cells to the damage site [108,109]. Thus, astrocytes can contribute to the development of dysfunction in the neurovascular unit because of their role in regulating blood flow, the extracellular balance of ions, and maintenance of the blood–brain barrier’s (BBB) functionality [110,111].

Microglia, in particular, have been singled out as major players in the progression of AD pathogenesis via neuroinflammation. Maintenance of a healthy CNS requires that microglia have a balance between two different phenotypes: M2, a state that has pro-angiogenic function and secretes anti-inflammatory cytokines for phagocytosis and clearing of cellular debris, and M1, a state that promotes inflammation [112,113]. Overactivation of the M1 phenotype can parallel AD progression. Transcriptome studies have shown that homeostatic genes are downregulated and AD-risk-factor genes, such as *APOE*, are upregulated in the transition of the microglial phenotype from protective to detrimental [114]. This transition could be modulated by a receptor expressed on myeloid cell 2 (TREM2) [114]. However, TREM2 knock-out experiments have yet to show conclusive results in terms of alleviating amyloid plaques and AD-related neurodegeneration. Furthermore, many studies have demonstrated that microglia could exhibit an incredibly diverse array of transcriptomic profiles depending on the stage of AD [115] and the brain region [116]. Thus, while microglial phenotypes along a pro-inflammatory-anti-inflammatory spectrum were speculated to contribute to AD progression, more research is needed to make sense of its temporal and spatial heterogeneity.

The interaction between microglia and the two most prominent hallmarks of AD, amyloid plaque and NFTs, deserves substantial attention. Microglia recognize Aβ through its binding to PRRs, especially TLRs, which triggers both the phagocytic and inflammatory pathways. While the former allows microglia to internalize and degrade Aβ [117], the latter consists of activated nuclear factor-[kappa]B (NF-[kappa]B), causing pro-inflammatory cytokines to be released and assembled into a pyrin domain-containing 3 (NLRP3) inflammasome [118,119]. The cytokines (e.g., IL-1β, IL-8, TNF) can activate more glial cells, damage more neurons, and even upregulate β-secretase, an enzyme that plays a crucial role in Aβ production (Figure 1) [120]. Similarly, NLRP3 inflammasomes can worsen amyloid pathology by exocytosing apoptosis-associated speck-like protein containing a C-terminal caspase recruitment domain (ASC), which might serve as a core for Aβ accumulation [121]. Exposure to tau proteins can change microglia morphologically, interfering with its function, and can even cause microglia to further disseminate tau through a phagocytosis and exocytosis mechanism (i.e., similar to that with Aβ) [122]. Thus, microglia are capable of propagating both Aβ and tau pathologies.

The relationship between the AD trajectory and microglia-mediated inflammation is rather complex. While microglia can be activated to damage the CNS, some evidence has suggested that microglia activation might have a protective role at first, which could be due, in part, to microglia’s ability to phagocytose and break down Aβ [117]. One study on patients in both prodromal and dementia stages of AD found that over 2 years, subjects with higher microglia tracer binding experienced slower cognitive decline [123]. However, a meta-analysis of compiled imaging data from 2019 revealed that neuroinflammation became more severe as patients progressed to later dementia stages of Alzheimer’s, which correlated with a drop in scores on the Mini-Mental State Examination. The spatial pattern of the spread of neuroinflammation in Alzheimer’s progression was also found to parallel the spatial evolution of tau [124]. The accumulation of Aβ and tau over time, coupled with neuroinflammation, which might have naturally built up due to aging, could eventually disrupt the microglial defensive mechanisms and prime the microglia towards inflammatory phenotypes and pathways [101].

Genetics have also been implicated in making an individual’s microglia more prone to pro-inflammatory dysfunction (Table 1). Several innate genes related to immune function have been identified to increase vulnerability to neurodegenerative disorders [125]. In the case of AD, the *APOE* gene, specifically the *APOE4* allele (Table 1), was deemed to be a major risk factor [126]. The APoE protein is the main transporter of apolipoprotein lipids and cholesterol in the CNS [127]. APoE can advance AD progression by not only influencing amyloid and tau pathology, but also disturbing microglia and astrocytes’ immunoprotective roles [125]. As discussed previously, the *APOE* gene was upregulated in disease-associated microglia (DAM) phenotypes [114], and one study found that in human induced pluripotent stem cells (iPSCs), going from the *APOE3* allele to the *APOE4* allele converted the microglial phenotype to DAM [128], which looked similar to the transcriptomic profile observed in the microglia of a human AD brain [129]. Astrocytes help to maintain the BBB and regulate lipid and glucose metabolism in the nervous system [125]. Exposure to pathogenic stimuli such as AB plaque or hyper-phosphorylated tau can trigger epigenetic alterations in astrocytes. Different stimuli can lead to different combinations of expressed genes, giving rise to a wide range of reactive astrocytes, some of which produce neurotoxic effects [129]. It is unknown whether the AD pathological pathway entails a particular signature of astrogliosis, but it is theorized that APoE could set off these potentially detrimental phenotype changes [125]. APoE plays an important part in lipid metabolism, and lipid accumulation in glial cells is a known hallmark of AD [125]. A 2021 study by Sienski et al. showed that, compared to iPSC-derived astrocytes carrying the *APOE3* allele, iPSC-derived astrocytes carrying the *APOE4* allele had greater amounts of lipid droplets and unsaturated fatty acids [130]. Moreover, *APOE4* glial cells were found to have excessive unesterified cholesterol, promoting the secretion of inflammatory cytokines and chemokines [131]. By exacerbating neuroinflammation, the APOE gene can greatly augment the risk of neurodegeneration.

Given that a host of evidence now shows neuroinflammation to be not just a side effect, but a crucial cause, of neurodegeneration, treatments designed to target inflammatory mechanisms are direly needed. PBM with transcranial NIR light is a non-invasive, non-pharmaceutical intervention that has risen in prominence in recent years for helping to reduce the glial reactivity and, therefore, neuroinflammation brought about by injury to the nervous system. Ma et al., in 2022, attempted to elucidate the mechanism by which PBM reduces the polarization of macrophages’ phenotypes to the neurotoxic M1 state while preserving the macrophages in the protective M2 state in mice with spinal cord injury (SCI), ultimately resulting in better recovery of motor function [132]. Prior models of SCI demonstrated most of the macrophages to be in the M1 state, expressing high levels of pro-inflammatory cytokines such as IL-6, IL-1α, and IL-1β [132,133]. Two major factors associated with M1 activation, hypoxia-inducible factor 1 (HIF-1) alpha and NF-kB, are stimulated by a mammalian receptor, Notch 1, in the Notch signaling pathway [132]. The same group found that PBM could inhibit the Notch1-HIF-1α/NF-kB axis-regulated polarization towards the M1 state and the release of inflammatory mediators, allowing for improvement of SCI symptoms [132].

Other studies have corroborated PBM’s promising effects on attenuating neuroinflammation (Figure 2). In 2016, Lu et al. demonstrated that low-level laser irradiation suppressed Aβ-induced microglia; astrocyte activation; and the production of IL-6, IL-1β, and TNFa inflammatory cytokines in the hippocampal regions of mice [134]. Many current drugs for AD were originally designed to target amyloid plaque aggregation, and perhaps a considerable reason for their inefficacy is that they were administered when Aβ levels in the brain had been elevated for far too many years. More and more research in Alzheimer’s pathology suggests that focusing on other factors linked with amyloid accumulation, neuroinflammation being one of the primary ones, can provide a superior avenue for slowing down neurodegeneration [134]. PBM fits exceptionally well into this criteria for a novel method that can produce better results in the treatment of AD.

### 2.3. Contribution of Glymphatic System

In the brain, neurons are supported by two extracellular fluids: interstitial fluid (ISF), which combines with the extracellular matrix to form the interstitial system between blood vessels and neural networks [135], and the CSF, which is produced in the choroid plexus and fills the cerebral ventricles to provide mechanical cushioning and maintain homeostasis for the brain [136]. CSF in the subarachnoid space flows into the brain interstitium via periarterial channels, where it can merge with ISF. CSF-ISF fluid then travels along perivenous pathways into the cervical lymphatic vessels for drainage. This forms the glymphatic system, a newly described structure primarily responsible for clearing out waste in the brain, but also partakes in lipid transportation and boosting glial signaling [137,138]. The glymphatic system is so named due to its reliance on glial cells—specifically, astrocytes—and its functional similarity to the lymphatic system [139]. Astrocytes regulate fluid transport between the interstitium and perivascular glymphatic spaces through water channels called aquaporin-4 (AQP4) [137].

The importance of AQP4 to the glymphatic system’s function cannot be understated. Aquaporins consist of six transmembrane helices, and AQP4 is the most common water channel in the brain [140]. By managing ionic and osmotic homeostasis, AQP4 influences the concentration gradients and diffusion of solutes necessary for neural function [139]. These water channels exist on the astrocyte end-feet (i.e., polarized distribution), allowing for CSF and ISF mixing and fluid exchange between perivascular spaces and the astrocyte cytoplasm and facilitating the waste-clearing drainage carried out by the glymphatic system [141]. AQP4 is one of the few markers expressed in all astrocytes, and animals lacking AQP4 have lowered solute clearance from the interstitial tissue, implying that the glymphatic system heavily depends on AQP4 [137,142].

Changes to AQP4 localization and expression lead to glymphatic system dysfunction, which can, in turn, further the development of neurodegenerative disorders. The glymphatic system has been hypothesized to help to clear out Aβ, as evidenced by observations such as the fact that Aβ clearance dropped by 50% in AQP4-knockout mice [137]. Zeppenfeld et al. in 2017 found that in the frontal cortex, enhanced AQP4 expression is a feature of the aging human brain, and its mislocalization from astrocyte end-feet (i.e., AQP4 depolarization) can render the brain more prone to harmful protein aggregations [143]. Other research has suggested that AQP4 depolarization is more of a consequence than a driving factor of Aβ accumulation [144]. Although the relationship between alterations to AQP4 and Aβ in AD pathology is still unclear, given the results of current studies, it is plausible that the initial buildup of Aβ can trigger loss of AQP4 localization, compounding the detrimental protein accumulation that has already occurred. Tau is another protein that can spread throughout the brain and assemble into NFTs. Some studies have observed that tau can use the extracellular space as a channel to jump from neuron to neuron, similar to prions [145], so any hindrance to the glymphatic system that translates to less effective clearance of extracellular space fluid could increase the risk of tau accumulating to dangerous levels. Research reported that in the AD mice model, brain regions such as the rostral and caudal cortex that exhibited greater tau deposition also experienced less CSF-ISF exchange and more AQP4 depolarization [146]. Adding TGN-020, an AQP4 inhibitor, led to a significant reduction in CSF-ISF flow and tau clearance, once again highlighting the essential nature of AQP4 in glymphatic function [146].

Neuroinflammation is known to exacerbate glymphatic drainage, and vice versa (Figure 1). The reactive gliosis seen in microglia and astrocytes accompanying neuroinflammation can cause the glymphatic flow to slow down [147]. A decrease in waste clearance via the glymphatic system can cause inflammatory mediators, such as cytokines and abnormal protein aggregations, to build up excessively, causing increased chronic inflammation. For instance, interference of the flow through the meningeal lymphatic vessels (MLVs) hindered drainage into the cervical lymph nodes, and was able to promote Aβ deposition in the brain meninges of mouse AD models [148]. The interaction between the immune system and the glymphatic system in AD pathology plays a unique role and deserves further investigation. A study by Feng et al. in 2020 attempted to explore the magnitude of the glymphatic system’s contribution to brain waste clearance relative to other management mechanisms such as enzymatic breakdown, phagocytosis by glial cells, and the BBB [149]. By using mice expressing APP and presenilin-1 (PS1) to increase Aβ production as a model for AD, they found that deleting the gene encoding for AQP4 resulted in a significant decrease in glymphatic clearance and extracellular accumulation of Aβ, but not this was severe enough to reach the point of Aβ plaque formation. Plaque deposition did occur in AQP4 knockout APP/PS1 mice when they selectively eliminated microglial function by injecting clodronate liposomes, but was not observed in APP/PS1 mice with functional AQP4. Taken together, the results indicated that the glymphatic system and microglia work synergistically to clear out Aβ and prevent Aβ plaque formation [149].

Susceptibility to AD appears to rise according to glymphatic system malfunction. PBM was able to augment glymphatic activity [150]. A 2019 study by Zinchenko et al. compared the Aβ-clearing effects of PBM at different power levels, finding that a skull fluence of 32 J/m^2^ significantly reduced Aβ accumulation without adverse side effects [151]. One possible explanation for these results is PBM’s ability to dilate MLVs, allowing for greater drainage of waste products such as Aβ. Research has suggested that PBM may induce relaxation and expand the diameter of extracranial lymphatic blood vessels, which corresponds with an increased clearance of markers such as gold nanorods (GNRs) [152]. Taking a deeper look at how PBM works at the cellular level, we can see that this vasodilation (Figure 2) likely occurs due to a PBM-induced increase in nitrous oxide via photodissociation of NO from mitochondrial CCO and activation of endothelial NO synthase (eNOS), which results in helping to maintain the vessel diameter to allow for sufficient perfusion [54,153]. PBM can also enhance MLV permeability by decreasing the expression of tight junction (TJ) proteins [154]. Not only does this allow for more fluid movement, it also allows more macrophages to enter into lymphatic vessels and break down Aβ [152]. This is supported by studies which have found that transcranial NIR light stimulation could reverse the harmful effects of AB-obstructed ISF flow [155]. An interesting observation was that PBM might not need to cover the entire brain to boost glymphatic system function. Thus, local intranasal applications of PBM were effective by minimizing blockage of the cribriform–lymphatic drainage route, a critical pathway for CSF outflow and primary bulk flow drainage from the brain [156]. PBM’s capacity to buttress the brain’s waste clearance system provides yet another explanation for its potential as a treatment for neurodegenerative disorders.

### 2.4. Role of Cerebral Blood Flow

Despite only weighing 2% of one’s total body weight, the human brain consumes around 20% of the body’s oxygen (O_2_) and glucose [157]. Within the cerebral vasculature, the arteries that feed into the brain snake throughout the cerebral surface and dive deeper into the parenchyma, branching into smaller arterioles and eventually capillaries. Cerebral blood flow (CBF) through the arteries, arterioles, and capillaries transports the necessary oxygen, glucose, and nutrients to the brain tissue to compensate for the demand. Circulation continues through veins which remove carbon dioxide (CO_2_) and waste products [157]. The mammalian brain can increase the rate of CBF in more active brain regions via a mechanism called neurovascular coupling or functional hyperemia [158]. A variety of cell types, including astrocytes, mural cells (i.e., vascular smooth muscle cells (VSMCs) and pericytes), and endothelial cells, form the neurovascular unit (NVU) and affect neurovascular coupling [158]. The structural composition of the NVU changes according to the different types of blood vessels. At the level of the penetrating arteries, the NVU consists of several endothelial cells lining the inner layer of the vessel wall and one to three layers of VSMCs. As the blood vessels transition to arterioles, the VSMCs only take up one layer. The capillaries, being the smallest type of blood vessel, barely have a layer of smooth muscle, but contain a greater presence of pericytes. The mural cells are largely responsible for regulating vessel diameter and blood flow at all points in the vascular tree [157]. It has been reported that Aβ deposited in the vessels could obstruct the NVU anatomical structure and disrupt the physiology of the unit, causing amyloid angiopathy. This disruption of fluid exchange and capillary flow in the brain causes local alterations and triggers neuroinflammation, ischemia, and local hypoperfusion of the brain, contributing to the mechanism of neurodegeneration linked to AD [157]. This hypothesis is supported by some researchers, and is called the neurovascular hypothesis of AD [159,160].

Several studies have suggested that decreased CBF and metabolism begin in preclinical AD, and could even precede Aβ or tau deposition (Figure 1) [161,162]. This drop is thought to be due to more than just tissue atrophy from Alzheimer’s, as the documented metabolic decrease exceeds the amount expected based on the degree of tissue atrophy [163], and AD brains tend to exhibit overly-constricted capillaries [164]. Much research has explored the latter factor, and the extreme narrowing has been attributed to dysfunction of the pericytes, the contractile mural cells that are most prominent around cerebral capillaries. When the pericytes contract excessively and narrow the capillaries, the blood flowing through flows more slowly than through capillaries with relaxed pericytes. This leads to a difference in capillary transit times (capillary transit time heterogeneity, or CTTH). A higher CTTH correlates with cognitive decline, as assessed by the Brief Cognitive Status Examination [165]. A 2019 study conducted by Nortley et al. found that in a human brain with Aβ, the capillary diameter increased quickly with greater distance from the pericyte soma, where the pericyte exerted its contractile effects. The rapid rate of increase demonstrated that capillary diameter is very susceptible to the presence of Aβ, implying that CBF can start to decrease well before reaching the level of Aβ accumulation which we observe in AD brains. Interestingly, the same study found that in human brains without Aβ, the capillary diameter decreased with greater distance from the pericyte soma [166].

The mechanism of capillary constriction induced by pericytes is still unclear. Current research efforts support a pathway by which Aβ oligomers trigger the release of ROS and endothelin-1 (ET-1), both of which are elevated in AD brains [167]. While some studies have pointed to perivascular macrophages as the sources of ROS [168], other studies have found that ROS are produced by microglia and pericytes [166]. ET-1 is known to activate endothelin A (ETA) receptors on pericytes, and although the locus of ET-1 generation was not determined, its elevated levels were consistent with the increased activation of ETA receptors in post mortem AD brains [169]. ROS and ETA receptor activation led to the narrowing of the capillaries, decreased CBF, and neurovascular uncoupling [167].

Neuroinflammation is another major mechanism contributing to CBF decline. In cerebral ischemia models, IL-1β upregulates ET-1, causing vasoconstriction [170]. This can easily be applied to AD models, as Aβ is known to increase the production of pro-inflammatory cytokines such as IL-1β from microglia and astrocyte inflammasomes. Additionally, neuroinflammation can increase the number of neutrophils and clotting factors [167]. In APP/PS1 mice, neutrophil aggregation can plug up capillaries and stall blood flow [171]. Researchers have shown that this blockage can be significantly alleviated by administering antibodies to the lymphocytes antigen 6 complex locus G6D (Ly6G), a neutrophil-specific marker. Inhibiting the Ly6G signaling decreased the number of neutrophils migrating to sites of inflammation [171]. Other studies demonstrated that an oral anticoagulant, dabigatran, could help to maintain a healthy level of CBF and slow down cognitive decline in AD mice, possibly by preventing excess fibrin buildup [172]. Furthermore, neuroinflammation can disrupt the function of the BBB, a specialized structure that regulates the exchange of materials between blood and brain tissue. Overactivation of glial cells generates excess proinflammatory mediators and ROS, which can alter the expression of tight junction proteins that seal the gaps between endothelial cells in the brain’s blood vessels, weakening the BBB’s ability to stop harmful molecules, such as more inflammatory cytokines, from crossing over into neural tissue [173]. AD mice models also exhibited increased production of matrix metalloproteinases (MMPs), enzymes that can degrade the extracellular matrix around blood vessels, and the tight junction proteins that help to form the BBB, induced by Aβ oligomers. Breakdown of the blood–CSF barrier in mice exposed to Aβ did not occur when MMP inhibitors were introduced or when the MMP-3 gene was knocked out [174].

The various mechanisms that impair CBF and neurovascular coupling can accelerate AD pathogenesis. Capillary constriction creates a hypoxic state, which can upregulate β-site amyloid precursor protein cleaving enzyme 1 (BACE1), the enzyme responsible for cleaving APP into neurotoxic, amyloidogenic Aβ isoforms. Gene sequence analysis has revealed that the BACE1 promoter region contains a hypoxia-responsive element (HRE). In conditions with low oxygen, HIF-1 binds to the HRE and increases BACE1 expression [175]. This mechanism offers a plausible pathway for decreased CBF possibly occurring before Aβ deposition: factors such as vascular defects, brain injuries, and genetic risk factors can hinder cerebral blood flow, inducing hypoxia and causing more Aβ to be produced by BACE1. The increased Aβ as a result of poor CBF can activate pericytes to contract and further narrow the cerebral capillaries, forming a positive feedback loop that exacerbates blood flow throughout the brain [167]. Capillary constriction could additionally provide a link between amyloid and tau pathology [167]. Cyclin-dependent kinase 5 (Cdk5) and glycogen synthase kinase 3 (GSK3) are major enzymes that hyperphosphorylate tau and form NFTs. Hypoxia inhibits the transport of calcium ions out of the cell, and the raised intracellular calcium concentration causes calpain to cleave Cdk5’s p35 regulatory subunit, enhancing Cdk5’s activity [176]. GSK3 is also activated by hypoxia through the downregulation of the phosphatidylinositol 3-kinase/Akt pathway [177].

Several interventions have been evaluated for their potential to treat AD by boosting CBF. PBM has emerged as a cutting-edge treatment for improving CBF. A 2018 study found that in patients with mild cognitive impairment, PBM applied to the areas of the vertebral arteries and internal carotid arteries, the major arteries providing blood to the brain, resulted in significantly greater CBF in the medial prefrontal cortex, lateral prefrontal cortex, anterior cingulate cortex, and occipital lateral cortex [178]. These increases in blood flow corresponded with better executive function and memory [178]. NIR light stimulations have been shown to augment CBF by raising levels of nitric oxide, an essential mediator of vascular homeostasis, through three mechanisms. First, PBM can activate Akt kinase to phosphorylate and upregulate endothelial nitric oxide synthase to generate more NO [179]. Second, PBM is capable of inducing the photodissociation of NO from mitochondrial cytochrome c oxidase directly [180]. Third, PBM can increase NO bioavailability from intracellular stores, including heme proteins such as hemoglobin and myoglobin [181]. More recent studies have demonstrated that low-level light administered in the 1000–1700 nm range could produce similar effects in phosphorylating Akt kinase, signaling it to translocate to specific sites in the cell where it could activate eNOS [182]. The range of 1000–1700 nm offers numerous advantages over lower wavelength ranges, including reduced light scattering, less absorption by tissue, and the ability to penetrate deeper regions of the brain at the same intensity [182]. By elevating NO levels, PBM can remedy poor circulation throughout the brain, consequently addressing a key factor in AD pathology (Figure 2).

Along with facilitating vasodilation through nitric oxide, PBM can also promote angiogenesis. In models of wound healing in the skin, PBM has been shown to increase angiogenesis, particularly through the stimulating expression of vascular endothelial growth factor (VEGF) [183]. Low-intensity light could also decrease the levels of MMP enzymes [183]. In normal conditions, MMPs are crucial in maintaining the extracellular matrix by degrading ECM proteins to balance ECM protein deposition. However, when tissue is damaged and in need of reconstruction, decreasing protein breakdown could be beneficial and could lead to a better environment for angiogenesis. Finally, an increase in hypoxia-inducible factor 1 was also found to be caused by PBM, and could lead to angiogenesis [183]. More work should be conducted to achieve the same results in the brain and elucidate the molecular mechanisms through which cerebral angiogenesis can occur by NIR light stimulation.

## 3. Discussion

Since its first documented case in the early 1900s, AD has emerged as the leading cause of dementia in the world. The most widely accepted model of Alzheimer’s pathogenesis, known as the amyloid-beta hypothesis, is based on the excessive accumulation of two proteins, amyloid-beta and tau, both of which are acknowledged as the signature hallmarks of AD brains [8]. This hypothesis posits that a mutation in the gene coding for amyloid precursor protein leads to the amyloidogenic pathway, where APP is first cleaved by β-secretase and then by γ-secretase, resulting in greater production of the insoluble and plaque-inducing Aβ1-42 isoform [8]. Amyloid plaque buildup consequently triggers downstream activation of kinase enzymes to hyperphosphorylate tau, destabilizing the protein binding to microtubules and causing large tau filaments to aggregate into insoluble intracellular NFTs [7]. Both Aβ and NFTs contribute to poor neural signaling and cell death [7].

The amyloid hypothesis has driven a substantial portion of AD clinical trials to be aimed at decreasing levels of Aβ plaque. In 2021, the US Food and Drug Administration (FDA) approved a drug called Aducanumab which was developed by the biotechnology company Biogen [184]. Aducanumab, an intravenously administered antibody, was shown to be able to reduce amyloid plaque in the brain and is the first approved drug that targets a root cause of AD pathology as opposed to temporarily alleviating symptoms [184]. The reasoning behind the FDA’s decision to approve this groundbreaking drug is rather nebulous. Two randomized phase 3 clinical trials, EMERGE and ENGAGE, were conducted to evaluate the efficacy of Aducanumab in early AD. Biogen stopped the trials early due to discouraging results, but after closer inspection, they observed that participants given the highest dose of Aducanumab in the EMERGE trial demonstrated statistically significantly slower cognitive decline. Participants in the ENGAGE trial demonstrated no statistically significant differences at any doses. Despite the ambiguity of the phase 3 studies, many professionals in the field of Alzheimer’s research support the FDA’s decision to approve this groundbreaking drug [185]. A study conducted in 2016 showed that Aducanumab could bind to Aβ plaque in the brain parenchyma and reduce both soluble and insoluble forms of Aβ in transgenic mice models. The study additionally administered Aducanumab monthly to human patients with AD for a year, and found that it reduced Aβ levels, which corresponded with slower cognitive decline as indicated by Clinical Dementia Rating—Sum of Boxes and Mini-Mental State Examination scores [186]. However, the FDA’s approval of Aducanumab was certainly met with skepticism. Past studies considering the effects of Aβ-targeting drugs have shown that they decrease the levels of amyloid plaque, but fail to alleviate Alzheimer’s symptoms [25,26,27,28,29], a sentiment echoed by several neurologists and biostatisticians. Many scientists have voiced their concern that Aducanumab’s approval will compel subsequent pharmacological development to similarly target Aβ plaque, stunting recent shifts to look at other factors of AD and design drugs to address them.

As the controversy around Aducanumab’s approval and what it means for Alzheimer’s research moving forward rages on, novel research devoted to exploring how other major factors drive AD pathology is becoming more important than ever. Neuroinflammation, as mentioned above, is an inflammatory response in the CNS that consists of pro-inflammatory cytokines, chemokines, small secondary messengers, and ROS (Figure 1). Chronically elevated levels of neuroinflammation can cause cell death and neurodegeneration via excessive amounts of cytokines and ROS [100]. Microglia and astrocytes are primarily responsible for mediating this inflammatory response, and Aβ and NFTs have been shown to over-stimulate the PRRs of these glial cells, resulting in detrimental cytokine storms and chemokine production. Exposure of microglia to Aβ additionally leads to NLRP3 inflammasome assembly. When the ASC domain of this inflammasome is secreted by microglia exocytosis, it can serve as a core for further Aβ accumulation [121]. Although much evidence suggests that Aβ provides a starting point for increased neuroinflammation, some studies have indicated that Aβ plaque and neuroinflammation do not always have a positive correlation, as neuroinflammation in the early stages of AD might even have a neuroprotective effect through microglia phagocytosis of Aβ [117]. One 2021 study assessed the correlations between CSF inflammatory markers and brain levels of Aβ and tau protein, revealing significant negative associations between Aβ and tau levels and inflammatory markers [187]. Their results were in line with previous longitudinal studies demonstrating that in prodromal AD and mild cognitive impairment (MCI) patients, higher levels of glial activation correlated with slower cognitive decline, and decreased glial activation was associated with more Aβ plaque. This not only implies that inflammatory factors function with a neuroprotective role for AD patients, but also possibly in cognitively normal older patients. In particular, they found that Aβ levels correlated negatively with IL-6 and IL-8 markers in the frontal brain regions, which are known to exhibit early Aβ accumulation. This matches the prior literature documenting IL-6′s ability to suppress amyloid deposition in mice models and IL-8’s angiogenic benefits in helping to clear Aβ across the BBB [187]. The study also observed negative correlations between tau protein levels and IL-8 and TNFα markers [187]. Overall, the relationship between pro-inflammatory cytokines and Aβ aggregation in AD pathology is incredibly complex, as the effect of inflammatory markers on progressing Alzheimer’s symptoms can vary depending on the stage of the disease.

Moreover, neuroinflammation is able to affect brain network function independently of amyloid plaque in AD brains. The disruption of structural and functional connectivity in the brain has been observed in AD and MCI brains [188]. A study conducted in 2022 used magnetic resonance imaging (MRI) to assess brain connectivity and relate it to Aβ deposition and microglial activation (i.e., neuroinflammation). Their findings suggested that microglial activation was negatively correlated with structural integrity and network local efficiency independently of Aβ levels, while network integrity was positively associated with cognition. Furthermore, Aβ was found not to be linearly correlated with changes in brain connectivity [188]. Previous evidence implies that the activated microglia of the M1 phenotype promote inflammatory conditions through ROS and cytokine generation, which damages neural axons and oligodendrocytes [112]. Activated microglia can additionally induce neuritic beading, i.e., bead-like swelling in dendrites and axons that is characteristic of neurodegenerative diseases, as well as mitochondrial dysfunction mediated through NMDA receptor signaling [189]. Overall, these results highlight a hypothesis for AD pathology, where neuroinflammation assumes a more central role and Aβ potentially does not considerably affect brain network integrity or cognitive decline once disease pathogenesis has begun.

Neuroinflammation’s significant role in neurodegeneration also underscores genetic risk factors for AD. Several genes related to innate immune function, of which the most well-known is the *APOE* gene, have been identified to increase vulnerability to neurodegenerative disorders [125]. The APOE protein is heavily involved in lipid metabolism, with microglia and astrocytes serving as the cells that express the highest amounts of APOE. However, APOE overproduction leads to an excessive inflammatory response [131]. Recent genome-wide association studies (GWAS) have identified upwards of 20 genes that can be potential risk factors for AD (Table 1), many of which regulate immune responses and encode proteins that are highly expressed in microglia [15,16,17]. Among them, *CD33*, *CLU*, *SORL1*, *CR1*, *MS4A*, *BIN1*, *CD2AP*, *PICALM*, *EPHA1*, *INPP5D*, *MEF2C*, *CASS4*, and *PTK2B* are most popular genes known to be associated with either immune response, which is related to microglia, or phagocytosis function, and are linked with AD pathogenesis at different ages of onset. Some AD-linked genes are associated with defects in cell migration, such as *CASS4* and *PTK2B* (related to the late onset of AD), or endocytosis and cytoskeletal organization malfunction, such as the *BIN1*, *CD2AP,* and *PICALM* genes (related to the late onset of disease) (Table 1). Along with the *APOE* gene, the *TREM2* gene arguably deserves the most attention (Table 1). TREM2 is a microglial surface receptor that has been implicated in modulating the transition from the protective M2 phenotype to the inflammatory M1 phenotype [114]. While some studies using transgenic mice models showed that TREM2 deficiency increased Aβ levels, other studies showed that TREM2 deficiency decreased Aβ levels, a discrepancy which could be attributed to the difference in the mouse models which were used [190]. Additional studies have suggested that mutations in the TREM2 gene leading to abnormal TREM2 receptors can disrupt microglial recruitment to plaque sites, inhibiting the phagocytosis and clearance of harmful Aβ protein [190]. Thus, conclusive results on TREM2′s effect in progressing AD pathology still have yet to be determined.

Oxidative stress constitutes another major contributor to exacerbating neuroinflammation. ROS are vital secondary messengers in cell processes regarding inflammation, and can activate microglia towards disease-associated microglia phenotypes via signaling pathways involving NF-kB and mitogen-activated protein kinases (MAPKs) [191]. Evidence suggests that hydrogen peroxide phosphorylates tyrosine residues on IκB kinase to activate NF-kB, and that ROS can activate a series of kinase enzymes to eventually activate MAPK [191]. NF-kB and MAPK can trigger downstream events resulting in the upregulation of cytokines and microglia activation. Activated microglia overproduce IL-1α, IL-1β, TNF-α, and even more ROS through overstimulation of NADPH oxidase (NOX), a group of seven ROS-generating enzymes that govern the oxidative burst in macrophages’ responses to pro-inflammatory signals [191]. In microglia, NOX2 and NOX4 are the most highly expressed transcripts [192]. When microglia are activated in AD pathology, the regulatory subunits of NOX2, p47phox, and p67phox are upregulated, which has been shown to correlate positively with oxidative stress and indicate NOX activation [193]. The insoluble Aβ-42 isoform induces NOX2 expression in the microglia as well, and this response is heightened in aged mice models [194]. Oxidative stress appears to feed into neuroinflammation and vice versa in a vicious cycle that only worsens AD progression.

Furthermore, several reports have demonstrated that Aβ plaque can increase oxidative stress. Amyloid beta is capable of forming complexes with metal ions such as copper, zinc, and iron, as well as generating superoxides and hydrogen peroxide [81,82]. On the other hand, oxidative stress can accumulate to dangerously high levels from lifestyle factors as well. The negative effects of oxidative stress compound over time as one ages [195]. Smoking, alcohol consumption, and a poor diet all contribute to oxidative stress. Physical exercise, especially moderate, low-intensity, and prolonged training, is known to be one of the best natural remedies for boosting antioxidant generation [196]. As a result, a sedentary lifestyle leads to higher levels of ROS. Furthermore, environmental factors including exposure to cigarette smoke, ultraviolet radiation, heavy metal ions, drugs or toxins, pollutants, and pesticides may all facilitate an increase in cellular ROS production [197]. ROS overproduction can then boost AB accumulation and aggravate neuroinflammation. It is, therefore, plausible to consider that Aβ plaque might not play a huge role in the progression of AD pathology. Age and one’s lifestyle habits and choices could elevate chronic oxidative stress and neuroinflammation to already harmful levels. Initial Aβ deposition could then provide the final straw which, upon exacerbating existing oxidative stress and inflammatory conditions, triggers a cascade of greater Aβ accumulation and worsening AD symptoms.

Neuroinflammation and oxidative stress combine and interact to damage neurons and bring about neurodegeneration. Another common characteristic of the AD brain is the inability to clear out detrimental molecules. Cerebral blood flows through the arteries, arterioles, and capillaries, supplying oxygen, glucose, and nutrients to the brain tissue. Circulation through the veins removes carbon dioxide (CO_2_) and waste products [157]. Impaired CBF through a capillary constriction is known to create hypoxic conditions, progressing AD pathology by upregulating β-secretase (BACE1 enzyme) and activating Cdk5 and GSK3 to hyperphosphorylate tau protein and form NFTs [167]. Capillary constriction is believed to occur through excessively contracting pericytes, which can be induced by Aβ deposition [167]. Neuroinflammation can cause further narrowing of the capillaries by disrupting the BBB and obstructing blood vessels with clotting factors [167]. However, underlying cardiovascular issues can also compromise cerebral vascular function. Atherosclerosis (stiffness and narrowing of the arteries due to plaque buildup on the inner wall), diabetes mellitus (prolonged high blood sugar), and midlife hypertension (high blood pressure) can all cause endothelial dysfunction and, subsequently, cerebral hypoperfusion [37]. Several studies have shown that midlife hypertension correlates with Alzheimer’s and dementia, while late-life hypertension does not exhibit as strong of an association [37]. Interestingly, the cardiovascular disease that is the easiest to connect with AD is one that does not affect the blood vessel anatomy directly: dyslipidemia, a condition where levels of high-density lipoproteins are reduced and levels of low-density lipoproteins are elevated. Carriers of the APOE4 allele tend to have higher cholesterol levels, which has been associated with increased Aβ production [37,49]. We can observe a disease model of AD wherein the brain’s capacity to prevent the buildup of neurotoxic chemicals is not only affected by Aβ plaque, but also through a host of other contributing factors, such as inflammatory response and cardiovascular conditions.

The glymphatic system is a recently discovered waste clearance system in the brain consisting of cerebrospinal fluid and interstitial fluid flowing through a network of microscopic channels and interstitial spaces around the blood vessels [139]. It is so named due to its dependence on glial cells and its functional similarity to the lymphatic system [139]. The glymphatic system has garnered substantial attention in the past several years due to its role in clearing out amyloid plaque, making it a promising drug target, especially for supporters of the amyloid hypothesis and because anti-Aβ plaque drugs are not proving to be as effective as expected [137]. However, the proposed discovery of the glymphatic system has sparked much debate in the scientific community. To date, no approved MR imaging technique has been able to definitively visualize the glymphatic system in the human brain [50]. Visualization techniques used in animal models, such as MRI tracer studies using an intrathecal injection of gadolinium-based contrast agent (GBCA), can pose threats if used in human patients. For example, gadolinium can induce encephalopathy along with losses of consciousness and seizures [50]. Outside of imaging difficulties, concerns surrounding the glymphatic system hypothesis persist. A 2016 study posited that the flow resistance through the AQP4 channels imperative to glymphatic function is too great to allow for the bulk transport of fluid [51]. Another study found that tracers injected into the CSF passed into the brain alongside arteries, but did not exit alongside veins as would be expected if the tracers flowed through the glymphatic system [198]. Instead, the tracers exited through periarterial basement membrane pathways, suggesting that there are other drainage systems in the brain [198]. Nevertheless, given the significant amount of literature demonstrating the glymphatic system’s ability to clear waste products in animal models and the CSF’s and ISF’s roles in maintaining brain homeostasis, future research could very much verify the glymphatic system’s importance. We must continue to refine and improve our current imaging techniques to continue investigating the glymphatic system’s contribution to AD pathogenesis and whether or not it is a suitable target for treatment interventions.

AD pathogenesis appears to be more complex than we ever thought. Models singling out excessive protein accumulation of amyloid beta and tau seem to be under fire and are possibly on the way to becoming obsolete, making room for models that include a variety of factors such as neuroinflammation, oxidative stress, and dysfunction in the CBF and brain clearance systems. Evidence suggests that these factors can progress AD pathology not only independently of Aβ, but also through interacting with Aβ and tau proteins to further aggravate neurodegeneration (Figure 1). Furthermore, all of these factors can magnify the effects produced by one another. Integrating all of this evidence leaves us with an incredibly layered and convoluted model that implies that the amyloid hypothesis does not provide a sufficient explanation of AD onset and progression. Instead, we must consider a holistic disease pathology model, where risk factors such as age, lack of physical exercise, a poor diet high in fat, smoking habits, genetics, and more can increase neuroinflammation and oxidative stress to above-average levels. Inflammatory conditions combined with detrimental lifestyle habits can lead to dysfunctional CBF and waste clearance networks, priming a patient’s brain to be more vulnerable to neurodegeneration. Aβ deposition then acts as an amplifier of the damage caused by all of the previously mentioned factors, serving as a tipping point as opposed to being the primary initiator of AD pathogenesis.

Thus, we need to explore treatments that can produce effects that are as varied as the factors that contribute to Alzheimer’s disease. Exposing cells to low-intensity light causes chromophores such as cytochrome c oxidase, a key enzyme in the mitochondrial ETC, to absorb photons. Light absorption can then trigger a series of biochemical reactions, including increased production of ATP, enhanced cellular metabolism, and increased release of ROS and photodissociation of nitric oxide [44]. The transient rise in ROS can subsequently activate NF-kB, which can serve as a transcription factor for genes modulating oxidative stress and neuroinflammation [96]. Another study by Ma et al. found that PBM was able to inhibit the Notch1-HIF-1α/NF-kB signaling axis, which activates microglia towards the M1 state and releases inflammatory mediators [132]. In addition, because NO is a vasodilator, PBM can boost glymphatic and cerebral blood flow [54,180].

Transcranial NIR light intervention has a variety of effects on the NF-kB pathway (Figure 2). While some studies using PBM have reported NF-kB activation [96], other studies have observed that PBM downregulates NF-kB in trauma and injury models [60]. Moreover, the activation of NF-kB, which contributes to pro-inflammatory pathways, by PBM seems counterintuitive to research demonstrating reduced neuroinflammation upon treatment with PBM [60,132]. These contradictions should be looked into and resolved in future experiments to better understand the molecular mechanism of PBM’s impact.

Overall, based on the current literature, we believe that PBM’s promising potential in treating AD is based on its ability to decrease harmful molecules not only through biochemical pathways, but also through increasing the brain’s capacity to clear detrimental compounds. This two-pronged attack on factors promoting neurodegeneration helps to explain why it offers advantages over other proposed interventions. Antioxidant therapies, for instance, have been proposed to alleviate increased oxidative stress, but they can only attempt to counteract the ROS that have already been formed. In comparison, PBM reduces ROS levels at the source of their production in the mitochondrial ETC, along with facilitating their clearance through the cerebral vasculature and glymphatic system. PBM should urge researchers to continue questioning traditionally accepted models of Alzheimer’s and pushing for treatments that address novel developments in AD pathogenesis.

## Figures and Tables

**Figure 1 ijms-24-09272-f001:**
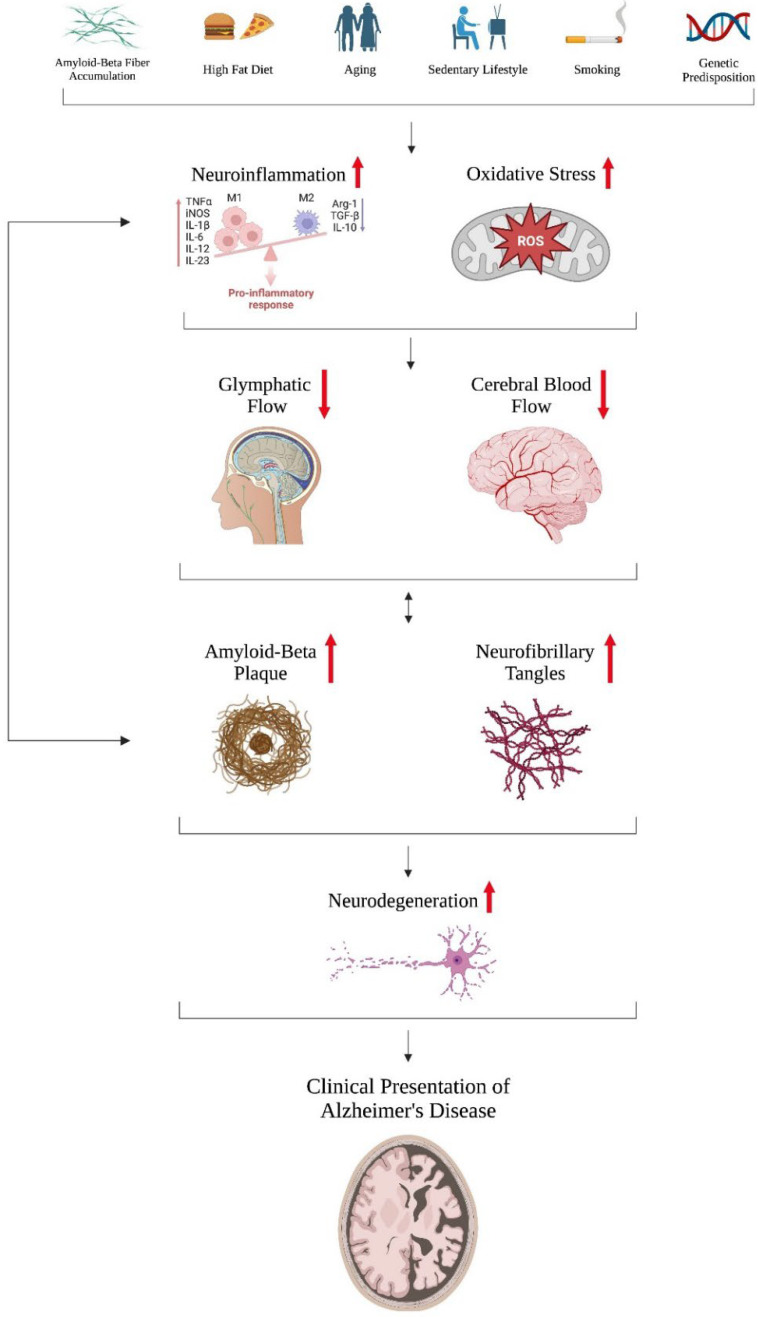
**Alzheimer’s disease (AD) pathogenesis**. This is a schematic presentation summarizing the series of events leading to the development of AD pathology and its clinical manifestation [8,9,10,15,16,22,27,32,34,37]. The figure was prepared using BioRender.com (accessed on 23 April 2023).

**Figure 2 ijms-24-09272-f002:**
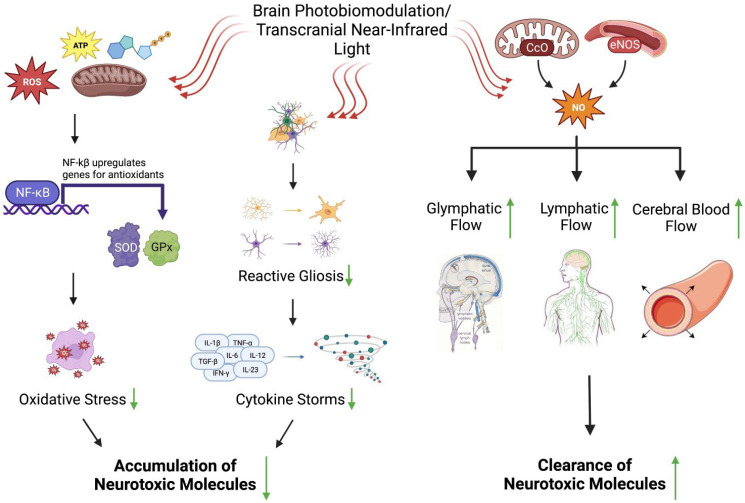
**Brain photobiomodulation in the treatment of Alzheimer’s disease (AD).** This is a schematic diagram of commonly known near-infrared light targets and the biological effects of transcranial photobiomodulation in the management of AD pathology and associated disease processes [44,45,47,48,49,50,51,52]. Abbreviations: ROS, reactive oxygen species; NO, nitric oxide; eNOS, endothelial nitric oxide synthase; SOD, superoxide dismutase; GPx, glutamate peroxidase; CcO, cytochrome C oxidase; TGF-β, transforming growth factor-beta; TNF-α, tumor necrosis factor-alpha; IFN-γ, interferon-gamma. The figure was prepared using BioRender.com (accessed on 25 April 2023).

**Table 1 ijms-24-09272-t001:** **Common genes linked to Alzheimer’s disease pathology.** This table summarizes the most common genes associated with AD. It also reflects the location of the genes, age of onset, the functions associated with AD genes, and effects of impaired/mutated genes in AD [15,16,17]. The familial form of AD (early-onset AD) is very rare (less than 1% of AD cases) and is caused by mutations in the *APP*, *PSEN1*, or *PSEN2* genes, with clinical onset of the disease before 60 years of age. The sporadic form of AD (late-onset AD or Mendelian inheritance) is the most common and reflects genetic risk factors and the environment, with clinical onset after 60 years of age. Mutations in *APOE*, *BIN1*, *Clu*, *CD33*, and *TREM2* are top-ranked genetic risk factors with the greatest probability of late-onset AD. All other genes listed in the table are risk factors indicating lower probability of developing the disease. Sporadic AD is associated with a dose-dependent increase in the risk of development and a decrease in age at the onset of AD. Abbreviations: Chr, chromosome; *PSEN1*, Presenilin 1; *PSEN2*, Presenilin 2; *ABCA7*, ATP-binding cassette transporter A7; *CD33*, cluster of differentiation 33; *TREM2*, triggering receptor expressed on myeloid cells 2; *Clu*, clusterin; *SORL1*, sortilin-related receptor; *CR1*, complement component (3b/4b) receptor 1; *MS4A*, membrane-spanning 4-domains A; *BIN1*, bridging integrator 1; *CD2AP*, CD2-associated protein; *PICALM*, phosphatidylinositol-binding clathrin assembly protein; *EPHA1*, EPH receptor A1; *NPP5D*, inositol polyphosphate-5-phosphatase D; *MEF2C*, myocyte enhancer factor 2C; *CASS4*, Cas scaffolding protein family member 4; *PTK2B*, protein tyrosine kinase 2 beta.

Gene	Chr	Age of Onset (Form of AD)	Gene Function Linked to AD	Effects of Gene Mutations on AD Pathogenesis
*APP*	21	Early/(Familial)	Aβ production, neural survival, synaptic function	Increases Aβ accumulation
*PSEN1*	14	Early/(Familial)	Aβ production, γ-secretase activity, calcium signaling	Increases Aβ42/Aβ40 ratio
*PSEN2*	1	Early/(Familial)	Aβ production, γ-secretase activity, synaptic plasticity	Increases Aβ42/Aβ40 ratio
*APOE*	19	Late/(Sporadic)	Lipid transport and metabolism, synaptic metabolism	Reduces Aβ clearance, increases Aβ deposition
*ABCA7*	9	Late/(Sporadic)	Phagocytosis, Aβ clearance	Increases Aβ accumulation
*CD33*	19	Late/(Sporadic)	Immune response, microglial phagocytosis	Increases Aβ accumulation, prevents Aβ from microglial clearance
*TREM2*	6	Late/(Sporadic)	Immune response, microglial phagocytosis	Reduces Aβ microglial clearance
*CLU*	8	Late/(Sporadic)	Immune modulation, cell death regulation	Increases Aβ aggregation, reduces Aβ clearance
*SORL1*	11	Late/(Sporadic)	Lipid metabolism, APP trafficking	Reduces Aβ destruction and endosomal degradation, impairs APP processing
*CR1*	1	Late/(Sporadic)	Immune response, Aβ clearance	Reduces Aβ clearance
*MS4A*	11	Late/(Sporadic)	Calcium signaling, immune function	Reduces sTREM2 and Aβ clearance
*BIN1*	2	Late/(Sporadic)	Endocytosis, cytoskeletal organization	Deregulates early endosome trafficking, induces neuron degeneration
*CD2AP*	6	Late/(Sporadic)	Endocytosis, cell signaling, cytoskeletal organization	Increases Aβ plaque formation, synaptic dysfunction, neurotoxicity
*PICALM*	11	Late/(Sporadic)	Endocytosis, synapse function, iron homeostasis	Increases Aβ production, reduces Aβ clearance, mediates tau neurodegeneration
*EPHA1*	7	Late/(Sporadic)	Immune response, synapse function, apoptosis	Promotes synaptic dysfunction and neurons apoptosis
*INPP5D*	2	Late/(Sporadic)	Immune regulation, cytokine signaling	Increases plaque deposition, promotes microglial and synaptic dysfunction
*MEF2C*	5	Late/(Sporadic)	Synapse construction, regulation of microglia	Mediates microglial overstimulation and increases neuroinflammation in late-onset
*CASS4*	20	Late/(Sporadic)	Cell migration and adhesion, inflammation	Increases NFT formation
*PTK2B*	8	Late/(Sporadic)	Cell migration, adhesion, and proliferation	Modulates tau pathogenesis, increases NFT formation

**Table 2 ijms-24-09272-t002:** **Common biomarkers linked to Alzheimer’s disease pathology.** This table summarizes the widely examined biomarkers associated with AD and reflects the different methods of pathogenesis [42,43]. Abbreviations: T-tau, total tau; P-tau, phosphorylated tau; NfL, neurofilament light; GFAP, glial cell line-derived neurotrophic factor.

Biomarker		Importance
**Amyloid biomarkers**	APP	APP cleavage by γ- and β-secretases results in Aβ formation
	Aβ42/Aβ40	Reduced Aβ42/Aβ40 ratio is observed in AD
**Tau biomarkers**	T-tau	Elevated in prodromal and dementia AD
	P-tau	Hyperphosphorylation of tau leads to NFT formation Elevated in prodromal and dementia AD High P-tau in CSF is only observed in AD
**Neural damage biomarkers**	NfL	Marker for acute brain damage and neurodegeneration, but not specific for AD
	S100β	High levels correlate with greater brain atrophy
**Neuroinflammation biomarkers**	GFAP	Marker of astrocyte activation Observed to be higher in preclinical AD cases
	TNF-α	Pro-inflammatory cytokine frequently reported to be elevated in blood plasma and CSF of AD patients
	IL-β	Promotes Aβ plaque and NFT formation
**Synaptic biomarkers**	α-synuclein	Elevated in CSF of MCI and AD patients
	Neurogranin	High neurogranin is observed in AD and reflects synaptic (dendritic) degeneration
**Metabolic biomarkers**	ApoE	Major lipid transporter in the brain May lead to synaptic defects and cognitive impairments
	GDNF	Promotes dopamine uptake in dopaminergic neurons Significant decrease reported in AD patients

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
