# Peer review of "Recent Mechanisms of Neurodegeneration and Photobiomodulation in the Context of Alzheimer’s Disease"

_ijms, 2023, doi:10.3390/ijms24119272_

Round 1

Reviewer 1 Report

Su et al. summarized the mechanisms of Alzheimer’s Disease pathogenesis. The authors especially pointed out the role of oxidative stress, neuroinflammation, the glymphatic system, and cerebral blood flow in AD pathogenesis. Furthermore, the review also summarized the mechanisms of photobiomodulation associated with AD pathology and the benefits of PBM treatment as of potential therapeutic solution.

There are plenty of reviews with partially overlapped topics published in the past several years in this area, including “Tian, Z., et al., Lasers in Medical Science, 2023”, “Monteiro, F., et al., Alzheimer's & Dementia: Translational Research & Clinical Interventions, 2022”, “Hamblin, M. R., Photonics, 2019” and more. However, the manuscript could still be meaningful due to the detailed mechanism analysis and plenty of references. To further improve the uniqueness of this review, the authors could highlight the most recent AD mechanism findings not discussed in previous reviews. However, the biggest issue for the manuscript is the lack of figures and tables, which make the review dull and unintuitive. I suggest the authors work out some figures and tables for further review.

1.       In the current version of the manuscript, there are no figures or tables. Adding some figures and tables could help the readers intuitively understand the molecular mechanisms of AD pathogenesis and how photobiomodulation could intervene the neurodegeneration. I will suggest adding at least one figure on biochemical pathways and molecular mechanisms of AD pathogenesis, another figure on the targets of PBM for AD therapy, a table to summarize the most important AD related biomarkers, genes, and signal molecules, and another table to summarize the approved AD drugs.

2.       The abbreviations for the word ‘photobiomodulation’ are not identical in the manuscript. Although it is ‘PBM’ in most parts of the manuscript, it appeared to be ‘PMD’ on lines 94 and 99 and ‘PPM’ on line 203. Please use consistent abbreviations for the same word and it’s better to add an abbreviation list to summarize all the abbreviations in the manuscript.

Author Response

We thank you for the time and detailed review of this work! Prior to addressing each reviewer’s comment with a point-by-point response we wanted to acknowledge the thorough review by you to help us to improve this communication. All reviewers noted no issues with the English language with only one remark recommending minor editing. This manuscript was edited by native speakers for readability and proofread for grammatical and other errors by Grammarly Premium Editing services. We take this seriously.

We also appreciate the positive feedback from reviewers, with the majority of ratings being above the average on the scale for the significance of this work to the field, scientifically sound organization and comprehensive context, adequate coverage with references, and well-written text. With the help of such a thorough review, we are making recommended changes and explaining our position on some raised topics to make it acceptable for publication.

Reviewer #1:

Comments and Suggestions for Authors

Su et al. summarized the mechanisms of Alzheimer’s Disease pathogenesis. The authors especially pointed out the role of oxidative stress, neuroinflammation, the glymphatic system, and cerebral blood flow in AD pathogenesis. Furthermore, the review also summarized the mechanisms of photobiomodulation associated with AD pathology and the benefits of PBM treatment as of potential therapeutic solution.

There are plenty of reviews with partially overlapped topics published in the past several years in this area, including “Tian, Z., et al., Lasers in Medical Science, 2023”, “Monteiro, F., et al., Alzheimer's & Dementia: Translational Research & Clinical Interventions, 2022”, “Hamblin, M. R., Photonics, 2019” and more. However, the manuscript could still be meaningful due to the detailed mechanism analysis and plenty of references. To further improve the uniqueness of this review, the authors could highlight the most recent AD mechanism findings not discussed in previous reviews. However, the biggest issue for the manuscript is the lack of figures and tables, which make the review dull and unintuitive. I suggest the authors work out some figures and tables for further review.

Response: Thank you for the valuable feedback! We agree with this reviewer and found this feedback to the point and constructive. This review focuses on the most progressive and recent scientifically backed views on mechanisms of neurodegeneration in the AD context. Even though some of the mechanisms had some coverage before, this review is unique and different compared to other literature sources in that it provides mechanisms of photobiomodulation in the context of AD pathogenesis by addressing the exact mechanisms of disease development using transcranial light therapy. We could also improve the readability of the manuscript and make it more intuitive by adding recommended figures and tables for consideration.

Some specific comments below:

  1. In the current version of the manuscript, there are no figures or tables. Adding some figures and tables could help the readers intuitively understand the molecular mechanisms of AD pathogenesis and how photobiomodulation could intervene the neurodegeneration. I will suggest adding at least one figure on biochemical pathways and molecular mechanisms of AD pathogenesis, another figure on the targets of PBM for AD therapy, a table to summarize the most important AD related biomarkers, genes, and signal molecules, and another table to summarize the approved AD drugs.

Response: We agree with the reviewer and as a result made Figure #1 with a schematic presentation of mechanisms of AD pathogenesis, Figure #2 with schematic targets and mechanisms of brain PBM in AD pathology. We also summarized the most common AD-linked biomarkers and genes in two tables and added them to the manuscript.

  1. The abbreviations for the word ‘photobiomodulation’ are not identical in the manuscript. Although it is ‘PBM’ in most parts of the manuscript, it appeared to be ‘PMD’ on lines 94 and 99 and ‘PPM’ on line 203. Please use consistent abbreviations for the same word and it’s better to add an abbreviation list to summarize all the abbreviations in the manuscript.

Response: We thank the reviewer for the detail orientation and corrected all abbreviated PBM typos throughout the entire text of the manuscript. Also, we added applicable abbreviations in figure legends and in the tables for the genes and biomarkers in the field.

Reviewer 2 Report

The manuscript developed by Su et al is interesting in the field of neurodegenerative diseases, including Alzheimer's disease, however, several modifications are required before being published in IJMS.

Major revisions

1. The introduction section is very long. I suggest adding other subsections that break down the neuropathological mechanisms associated with Aβ and Tau.

2. In the introduction, describe that the Aβ accumulation forms amyloid plaques and also can lead to amyloid angiopathy. In this context, mention the neurovascular hypothesis (doi: 10.1038/nrn3114; 10.3390/ijms22042022).

3. Add a subsection of PMD, which includes definitions, generalities, adverse effects, and contraindications. Emphasizing if there are clinical trials in patients with AD, what would be the suggested frequency and duration for AD? Would it influence the time of evolution of the patient? What are the limitations of the use of PMD? Are there studies that do not show favorable effects? Please discuss.

4. In the section “2. Mechanisms of pathogenesis of Alzheimer's disease” the information in the text is extensive, I suggest you summarize it in tables. In addition, as a review article, it would improve and be more attractive if the authors included a figure (s) illustrating the main mechanisms of neurodegeneration and the possible effects of photobiomodulation therapy in AD.

5. In the neuroinflammation subsection, some physiological roles of M2 remain to be described (https://doi.org/10.3390/cells11132091). Likewise, it is necessary to include more roles of astrocytes under physiological conditions (https://doi.org/10.1186/s40035-020-00221-2) and the phenotypes of A1 and A2 astrocytes (DOI: 10.1038/nature21029) and discuss the possible effect of PBM on these. In this same section, authors could describe the role of pyroptosis and whether there is evidence that PMD decreases it.

6. Change the focus of section 2.4 to “dysfunction of the neurovascular unit”. Please describe how insoluble Aβ would generate NVU dysfunction (DOI: 10.3390/ijms22073654).

7. Section 3 “discussion” is extensive and repetitive of what was mentioned in section 2. Integrate into a single section.

8. Around 50% of the references are before 2016, I suggest updating where applicable.

Minor revisions

1. It is a pleonasm "normal physiological", please remove the word normal.

2. Change the term from normal to the non-amyloidogenic pathway (L51) and diseased pathway to the amyloidogenic pathway (L556).

3. In L133, remove “treating hypertension” since it is understood that PBM is used to treat hypertension.

4. In Line 203 change PPM to PBM

5. Homogenize the term PMD or PBM.

6. Abbreviations are missing throughout the text.

Minor editing of English language required

Author Response

We thank you for the time and detailed review of this work! Prior to addressing each reviewer’s comment with a point-by-point response we wanted to acknowledge the thorough review by you to help us to improve this communication. All reviewers noted no issues with the English language with only one remark recommending minor editing. This manuscript was edited by native speakers for readability and proofread for grammatical and other errors by Grammarly Premium Editing services. We take this seriously.

We also appreciate the positive feedback from reviewers, with the majority of ratings being above the average on the scale for the significance of this work to the field, scientifically sound organization and comprehensive context, adequate coverage with references, and well-written text. With the help of such a thorough review, we are making recommended changes and explaining our position on some raised topics to make it acceptable for publication.

Reviewer #2:

Comments and Suggestions for Authors

The manuscript developed by Su et al is interesting in the field of neurodegenerative diseases, including Alzheimer's disease, however, several modifications are required before being published in IJMS.

Some specific comments below:

Major revisions

  1. The introduction section is very long. I suggest adding other subsections that break down the neuropathological mechanisms associated with Aβand Tau.

Response: Thank you for the suggestion. We agree that the introduction section seems lengthy. This is because the authors wanted to convey to readers available hypotheses, help to navigate from more popular but outdated ideas to more current, technologically advanced, and supported hypotheses. This takes a little more reading and contributes to the volume of the text and “older” references cited in this manuscript (please refer to the major revision point# 8).

  1. In the introduction, describe that the Aβ accumulation forms amyloid plaques and also can lead to amyloid angiopathy. In this context, mention the neurovascular hypothesis (doi: 10.1038/nrn3114; 10.3390/ijms22042022).

Response: We appreciate valuable comments and suggestions. The authors added a section and gave credit to the neurovascular hypothesis in this review in section 2.4 focused on the Role of Cerebral Blood Flow.

  1. Add a subsection of PMD, which includes definitions, generalities, adverse effects, and contraindications. Emphasizing if there are clinical trials in patients with AD, what would be the suggested frequency and duration for AD? Would it influence the time of evolution of the patient? What are the limitations of the use of PMD? Are there studies that do not show favorable effects? Please discuss.

Response: We thank the reviewer for the thoughtful recommendations! However, the scope and goal of the manuscript are outside of recommended changes. Even though PBM adds uniqueness to this review to make a positive contribution by aiming exact mechanisms of neurodegeneration pathogenesis causing AD and introducing a comprehensive way for readers’ awareness regarding evolving light technology, however, this review was not designed to address many variations of the treatment protocol, technological modifications of different devices, physical properties of the light, treatment stimulation approaches, inclusions and exclusions of enrollment, side effects, etc. Our group would be happy to share links to published resources and our findings using this technology and happy to share the results of our population in the clinical trial that did not have treatment-related side effects or adverse reactions. More research/clinical trials need to be done to discuss and review therapeutic protocol using an appropriate dosimetry approach, so it is interchangeable between different devices in the biomedical field. Please review our communication for the dosimetry reference: DOI: 10.3389/fphar.2022.965788.

  1. In the section “2. Mechanisms of pathogenesis of Alzheimer's disease” the information in the text is extensive, I suggest you summarize it in tables. In addition, as a review article, it would improve and be more attractive if the authors included a figure (s) illustrating the main mechanisms of neurodegeneration and the possible effects of photobiomodulation therapy in AD.

Response: We appreciate the reviewer’s suggestion and valuable advice! To address this point, we added figures and tables to improve the readability and schematic presentation of processes in AD pathogenesis and the summary of PBM effects in the AD context.

  1. In the neuroinflammation subsection, some physiological roles of M2 remain to be described (https://doi.org/10.3390/cells11132091). Likewise, it is necessary to include more roles of astrocytes under physiological conditions (https://doi.org/10.1186/s40035-020-00221-2) and the phenotypes of A1 and A2 astrocytes (DOI: 10.1038/nature21029) and discuss the possible effect of PBM on these. In this same section, authors could describe the role of pyroptosis and whether there is evidence that PMD decreases it.

Response: We thank the reviewer for the detailed suggestions. We have reviewed the suggested information and found that recommended Cells paper (added to the reference list and cited appropriately) would add beneficial information to readers regarding subtypes of M2 and all phenotypes in the physiology of the brain and important pro-angiogenic and anti-inflammatory cell protective functions. The same recommendation regarding astrocytes functionality was well accepted, and text was added to the manuscript with appropriate suggested references.

  1. Change the focus of section 2.4 to “dysfunction of the neurovascular unit”. Please describe how insoluble Aβwould generate NVU dysfunction (DOI: 10.3390/ijms22073654).

Response: Thank you for the feedback! After careful consideration, the authors decided to leave the name of subsection 2.4 as “Role of Cerebral Blood Flow”. However, a short description of the Neurovascular hypothesis was added with an explanation of how it leads to the development of amyloid angiopathy and its contribution to neurodegeneration in AD. Citations and references were also added to the text.

  1. Section 3 “discussion” is extensive and repetitive of what was mentioned in section 2. Integrate into a single section.

Response: We agree that discussion can be redundant to some extent but this “redundancy” is to keep the reader on track of discussed items for more focus.

  1. Around 50% of the references are before 2016, I suggest updating where applicable.

Response: Authors appreciate the reviewer’s comment and thorough evaluation of the manuscript. We also agree that AD pathology is an established disease state. Some of the information provided to the readers has established knowledge with background and introduction in this area of research therefore, some reports may have older date stamps. We also added several references to support the discussed ideas reviewers pointed out and ensured that research originated from recent years to address this point even further.

Minor revisions

  1. It is a pleonasm "normal physiological", please remove the word normal.

Response: Thank you for the careful evaluation of the wording! We agree with the reviewer and worded it appropriately.

  1. Change the term from normal to the non-amyloidogenic pathway (L51) and diseased pathway to the amyloidogenic pathway (L556).

Response: We agree with the reviewer and made appropriate changes to the text.

  1. In L133, remove “treating hypertension” since it is understood that PBM is used to treat hypertension.

Response: Authors agree with the reviewer, and the text was removed as suggested.

  1. In Line 203 change PPM to PBM

Response: Thank you for your thorough reading and comment. All PBM abbreviations, along with others, were corrected throughout the manuscript.

  1. Homogenize the term PMD or PBM.

Response: The text was homogenized and corrected.

  1. Abbreviations are missing throughout the text.

Response: The text of the manuscript was proofread and missing abbreviations with elaboration were added where applicable. This also applies to the figure and table legends.

Reviewer 3 Report

Authors reviewed on the topic of the mechanisms of neurodegeneration in AD with following, "Recent Mechanisms of Neurodegeneration and Photobiomodulation Therapy in the context of Alzheimer’s Disease"

Majority of the contents were in line with the mechanisms of neurodegeneration in AD and only discussed very minimum on the photobiomodulation without mentioning many significant references, especially on the magical frequency of 40 Hz. The title can mislead the readers on the contents. Hence, authors should revise the title to fit the contents by omitting the photobiomodulation or bring significant references on the photobiomodulations (and/or other sound, vibrational, vision, and radio therapies, including their mechanisms) into the manuscript.

Author Response

We thank you for the time and detailed review of this work! Prior to addressing each reviewer’s comment with a point-by-point response we wanted to acknowledge the thorough review by you to help us to improve this communication. All reviewers noted no issues with the English language with only one remark recommending minor editing. This manuscript was edited by native speakers for readability and proofread for grammatical and other errors by Grammarly Premium Editing services. We take this seriously.

We also appreciate the positive feedback from reviewers, with the majority of ratings being above the average on the scale for the significance of this work to the field, scientifically sound organization and comprehensive context, adequate coverage with references, and well-written text. With the help of such a thorough review, we are making recommended changes and explaining our position on some raised topics to make it acceptable for publication.

Reviewer #3:

Comments and Suggestions for Authors

Authors reviewed on the topic of the mechanisms of neurodegeneration in AD with following, "Recent Mechanisms of Neurodegeneration and Photobiomodulation Therapy in the context of Alzheimer’s Disease"

Majority of the contents were in line with the mechanisms of neurodegeneration in AD and only discussed very minimum on the photobiomodulation without mentioning many significant references, especially on the magical frequency of 40 Hz. The title can mislead the readers on the contents. Hence, authors should revise the title to fit the contents by omitting the photobiomodulation or bring significant references on the photobiomodulations (and/or other sound, vibrational, vision, and radio therapies, including their mechanisms) into the manuscript.

Response: We are very grateful for the important remark made regarding photobiomodulation and the title. Let us explain why authors would like to preserve the current integrity of the manuscript and why “magical frequency” with other frequencies and wavelengths deserves a separate contribution with a focus on PBM and treatment protocols and therefore outside of the scope of this review. The current title focuses readers on some mechanisms of pathogenesis of neurodegeneration and PBM mechanisms linked to discussed pathogenesis in the AD context. This review’s goal is to focus on the most recent findings only in AD-related evidence-based research supporting these mechanisms. As far as PBM, this field is relatively new with a limited amount of reports done in recent years. Reviewer noted that we discussed very minimum on PBM without mentioning many significant references. This manuscript has 39 references with a focus on PBM in support of discussed mechanisms; considering this new area of research, it is an extensive amount of references on this topic. Regarding the “magical frequency” of 40Hz or any other impulse frequency of the light, the wavelength of the light and treatment protocol is outside of this communication’s scope. Here is why: Our group has extensive experience and knowledge in the area of PBM. We have invented a helmet device, and successfully conducted two clinical trials using transcranial near-infrared light stimulations at the convenience of patients’ homes resulting in a few original publications: DOI: 10.14336/AD.2021.0229 ; DOI: 10.7759/cureus.16188 ; and several review papers: DOI: 10.3389/fphar.2022.965788 ; DOI: 10.4103/1673-5374.366499 on this topic of research. Considering that this technology is forgiving for experimentation (minimal to none side effects if used correctly) and the development of treatment protocols (for AD including) can provide hundreds of different combinations and can introduce a lot more confusion to the readers if not specifically addressing the goal of communication one at present time. Thus, light can be emitted on the surface of the head, introduced through the orbits or nasopharyngeal junction; it can be only frontal exposures or covering some areas of the brain projection or covering the whole brain. Light can illuminate at various wavelengths (from a visible spectrum of 700nm to a near-infrared light spectrum ranging from 800nm to 1300nm depending on the application). Treatment sessions can be used as a continuous light application (no impulse) or flickering (at impulses- most popular of them are at 10Hz and 40Hz). Effectiveness can differ depending on the chosen power of the light source, the intensity (very low, low, medium, or intense), the irradiance of the light, distance, the color of the hair, and the color of the skin. Outcomes will depend on what parameter you choose from above and what treatment protocol you will use it with- how long for each session, how many times a day, every day or not, how long for the treatment course, questions of dosimetry, etc. And these variables are endless when you combine them to establish a treatment protocol. What adds more flavor is what pathology you are trying to manage, how old your patient is, how old the disease state is and at what stage, what other comorbidities and ongoing treatment regimens/treatment plan the patients are on… By listing these variables, I would like to bring to your attention the amount of information regarding PBM treatment that deserves to be published without additional content and can be done in future communications.

Round 2

Reviewer 1 Report

I am so sorry but I did not see the new figures and tables in the new version. There might be some edition issue. Please resubmit for further review.

Author Response

We apologize for missing figures and tables in the submitted revised text. This was never intended. To our surprise, figures and tables uploaded on the portal during the submission process at the “Figure upload tab” were not correctly populated with the submission of the main file. To avoid this from happening, the authors attached recommended figures and tables at the end of the main body of the manuscript after the list of references. Thank you for your patience! All missing figures and tables should now be populated in the main body.

Reviewer 2 Report

The authors followed most of the recommendations and significantly improved the manuscript. However, the attached document does not show the figures and tables. To be published in IJMS it is necessary to attach them.

Minor editing of the English language required

Author Response

We highly value the reviewer’s feedback and appreciate the supportive nature and positive comments regarding changes made in the revised text of the manuscript and the justification used in response to comments. We apologize for missing figures and tables in the submitted revised text. To our surprise, figures, and tables uploaded on the portal during the revision submission process at the “Figure upload tab” were not correctly populated with the submission of the main file. To avoid this from happening, the authors attached recommended figures and tables at the end of the main body of the manuscript after the list of references.

The English language was evaluated more, and editing was done to address it further.

Reviewer 3 Report

Authors tried to revise the manuscript according to the previous comments and reasoned for their criteria for the PMT.

It's an interesting to subject, but the authors are bias towards to one avenue of PMT, which authors need to scope the various areas of PMT in the review manuscript.

Importantly, there as no figures and tables were in the manuscript to be reviewed properly.

In addition, author included many references at the end of the reference section with citing them in the text.

Hence, the carelessness of authors prompt the review of the manuscript to be process further.

Minor grammar mistakes on the verb tenses. If a sentence refers to a reference, the verb needs to be in the past tense. General knowledge of the fact can be in the present tense.

Author Response

We greatly appreciate the thorough review and detail-oriented approach of the reviewer!  The English language was evaluated more, and minor grammar mistakes pointed out by the reviewer were addressed further throughout the manuscript.

We apologize for missing figures and tables in the submitted revised text. To our surprise, figures, and tables uploaded on the portal during the revision submission process at the “Figure upload tab” were not correctly populated with the submission of the main file. To avoid this from happening, the authors attached recommended figures and tables at the end of the main body of the manuscript after the list of references.

Various areas of PBM effects in the context of AD neurodegeneration mechanisms were pointed out in each section/subsection of the manuscript, with further discussions in the discussion section. PBM is evolving, promising, and forgiving technology that can be used in various biomedical applications, including dementia and AD treatment. Our group will be happy to design a review with only a focus on PBM. As was stated in Revision 1, this area of research has much to offer. It can be confusing for readers if it is not focused because involves many parameters on the physical properties of the light, electromagnetic parameters, selected and justified treatment protocol for each disease state, duration of treatment and state of disease, and many more to cover. Please, extend an invitation to our group, and we will be happy to contribute with a quality publication focusing on the therapeutic side of the PBM spectrum.

Round 3

Reviewer 1 Report

The manuscript was dramatically improved by adding some figures and tables. However, there are still some issues that need further attention. 

1. Many genes and biomarkers in the tables are not mentioned in the main text at all. If these genes and biomarkers are really important in the mechanisms of neurodegeneration, they should be introduced with more details in the main text. Or they should not be included in the table to avoid any confusion.

2. Some abbreviations in the tables are not consistent with the main text. For example, Presenilin 1 is shown as PS1 in the main text but in Table 1 it is PSEN1. This is why I suggested having an abbreviation list to summarize all the abbreviations in the manuscript rather than listing them separately in figures and tables.

3. The tables and figures should be able to provide more information than the present form. For example, Table 1 should be able to list how the gene expression changed (up- or down-regulated) in AD patients. And Figure 2 should be able to illustrate the gene or biomarker targets of PBM for AD therapy. 

Due to the concerns listed above, I suggest the authors revise the manuscript again for further review.

Author Response

We thank you for the time and detailed review of this work! Before addressing each reviewer’s comment with a point-by-point response, we wanted to acknowledge your valuable contribution to improving this communication. Because of a combined effort working in orchestration with knowledgeable reviewers, this manuscript was significantly improved, which was pointed out by all reviewers in the previous revision cycle. This manuscript was edited by native speakers for readability and proofread by Grammarly Premium Editing Services. Thank you once again for your commitment and dedication!

We also appreciate the positive feedback from reviewers, with all five evaluation ratings improved throughout the revision process! With such a thorough review by the experts in the field, the authors improved the quality, impact, and readability of this manuscript and are hopeful that it is ready to be accepted for publication.

Comments and Suggestions for Authors:

The manuscript was dramatically improved by adding some figures and tables. However, there are still some issues that need further attention.

Response: We appreciate the reviewer’s constructive contribution and positive feedback regarding the quality of this communication! This comprehensive review has specific agenda to address, focusing on the most recent scientifically supported mechanisms of neurodegeneration associated with AD pathology. It discusses why mechanisms previously believed to cause AD is no longer as supported by the scientific community as in former years. This review also adds uniqueness by addressing the exact mechanisms of pathogenesis with scientifically supported effects of transcranial PBM implementation. This would help readers to stay focused on mechanisms and have PBM awareness. We appreciate the reviewer’s help in keeping it organized and focused.

  1. Many genes and biomarkers in the tables are not mentioned in the main text at all. If these genes and biomarkers are really important in the mechanisms of neurodegeneration, they should be introduced with more details in the main text. Or they should not be included in the table to avoid any confusion.

Response : Thank you for the valuable comment! We agree with the reviewer that biomarkers-related information falls slightly outside the manuscript’s scope and can overwhelm readers if included in this communication. However, we also agree that some biomarkers can be linked to AD pathology, and adding the commonly studied AD-associated genes and biomarkers in a concise way would benefit readers looking for the related information and answers in one spot in that area of AD research. Therefore, we introduced and added related context to the main body in reference to genes and biomarkers to address the reviewer's point.

  1. Some abbreviations in the tables are not consistent with the main text. For example, Presenilin 1 is shown as PS1 in the main text but in Table 1 it is PSEN1. This is why I suggested having an abbreviation list to summarize all the abbreviations in the manuscript rather than listing them separately in figures and tables.

Response: We understand the reviewer’s concern about the noted abbreviation. However, it is general knowledge for genes, proteins, and knock-out animal model names to have established names in the biomedical field. Therefore we can not modify common names for established models of knock-out animals to adopt a new abbreviation of choice in this manuscript. The same approach applies to commonly known genes like PSEN1 and PSEN2. The reviewer’s example with Presenilin 1 and 2 protein names (PS1 and PS2, respectively) are the only mismatched exceptions in this manuscript and therefore were included in the list of abbreviations in Table 1 legend to avoid a potential confusion by the reader.

  1. The tables and figures should be able to provide more information than the present form. For example, Table 1 should be able to list how the gene expression changed (up- or down-regulated) in AD patients. And Figure 2 should be able to illustrate the gene or biomarker targets of PBM for AD therapy.

Due to the concerns listed above, I suggest the authors revise the manuscript again for further review.

Response: This is a great point made by the reviewer regarding Table 1. We have added a suggested approach in gene expression levels similar to Table 2 for biomarkers and the associated effect of mutation involving each corresponding gene. Regarding Figure 2 comment to illustrate the gene or biomarker targets of PBM in AD would not be an accurate approach within the scope of this review because targets are expected to be different depending on the type of PBM and selected therapeutic protocol, which includes and are not limited to the type of light, wavelength, power of light, irradiance, frequency/pulse, duration, and location of brain area where the light is used. Based on our previous clinical AD research using PBM, we suggest that this topic is communicated in research focusing on treatment, therapeutic modalities, dosimetry, and other parameters affecting intracellular and extracellular light targets depending on the physical characteristics mentioned above.

Reviewer 2 Report

Good job!

 Minor editing of English language required

Author Response

Dear Reviewer,

We thank you for the time and detailed review of this work! Before addressing each reviewer’s comment with a point-by-point response, we wanted to acknowledge your valuable contribution to improving this communication. Because of a combined effort working in orchestration with knowledgeable reviewers, this manuscript was significantly improved, which was pointed out by all reviewers in the previous revision cycle. This manuscript was edited by native speakers for readability and proofread by Grammarly Premium Editing Services. Thank you once again for your commitment and dedication!

We also appreciate the positive feedback from reviewers, with all five evaluation ratings improved throughout the revision process! With such a thorough review by the experts in the field, the authors improved the quality, impact, and readability of this manuscript and are hopeful that it is ready to be accepted for publication.

Comments and Suggestions for Authors:

Good job!

Response: We appreciate the positive comment and encouraging feedback from the reviewer! Thank you for your contribution to this work!

Comments on the Quality of English Language:

Minor editing of English language required

Response: We thank the reviewer for the thorough evaluation!  Native speakers edited English as requested to satisfy the reviewer and improve manuscript readability.

Reviewer 3 Report

Authors made significant improvements with this manuscript.

However, it still missing citations of references in the text, which were listed in the list. The list is not in order. Authors must go through all references carefully and match each reference with the corresponding citation.

Grammar in verb tenses were wrongly used. The present tense can be used for the general knowledges and the past tense should be used to all cited references.

Author Response

Dear Reviewer,

We thank you for the time and detailed review of this work! Before addressing each reviewer’s comment with a point-by-point response, we wanted to acknowledge your valuable contribution to improving this communication. Because of a combined effort working in orchestration with knowledgeable reviewers, this manuscript was significantly improved, which was pointed out by all reviewers in the previous revision cycle. This manuscript was edited by native speakers for readability and proofread by Grammarly Premium Editing Services. Thank you once again for your commitment and dedication!

We also appreciate the positive feedback from reviewers, with all five evaluation ratings improved throughout the revision process! With such a thorough review by the experts in the field, the authors improved the quality, impact, and readability of this manuscript and are hopeful that it is ready to be accepted for publication.

Comments and Suggestions for Authors:

Authors made significant improvements with this manuscript.

Response: Authors are grateful for the reviewers’ dedication to improving this manuscript and thankful for given feedback regarding the achieved improvements of this work!

However, it still missing citations of references in the text, which were listed in the list. The list is not in order. Authors must go through all references carefully and match each reference with the corresponding citation.

Response: We greatly appreciate the reviewers’ detailed orientation! All references and citation orders were thoroughly checked and addressed as requested. This applies to Tables and Figures cited in the order they appear in the text. Thank you!

Comments on the Quality of English Language:

Grammar in verb tenses were wrongly used. The present tense can be used for the general knowledges and the past tense should be used to all cited references.

Response: We thank the reviewer for the thorough evaluation and feedback!  Native speakers edited the English as requested to satisfy the requirement, to improve manuscript readability, and to address the point regarding the time tense.

Round 4

Reviewer 1 Report

The author fully addressed my concerns. The manuscript has been improved a lot and should be good enough to publish on IJMS.